# Insect decline in forests depends on species' traits and may be mitigated by management

Michael Staab [1][✉], Martin M. Gossner [2,3], Nadja K. Simons[1], Rafael Achury[4], Didem Ambarlı[4,5], Soyeon Bae[4,6], Peter Schall [7], Wolfgang W. Weisser [4] & Nico Blüthgen [1]

Insects are declining, but the underlying drivers and differences in responses between species are still largely unclear. Despite the importance of forests, insect trends therein have received little attention. Using 10 years of standardized data (120,996 individuals; 1,805 species) from 140 sites in Germany, we show that declines occurred in most sites and species across trophic groups. In particular, declines (quantified as the correlation between year and the respective community response) were more consistent in sites with many non-native trees or a large amount of timber harvested before the onset of sampling. Correlations at the species level depended on species' life-history. Larger species, more abundant species, and species of higher trophic level declined most, while herbivores increased. This suggests potential shifts in food webs possibly affecting ecosystem functioning. A targeted management, including promoting more natural tree species composition and partially reduced harvesting, can contribute to mitigating declines.

[1] Ecological Networks, Technische Universität Darmstadt, Schnittspahnstraße 3, 64287 Darmstadt, Germany. [2] Forest Entomology, WSL Swiss Federal Research Institute, Zürcherstrasse 111, 8903 Birmensdorf, Switzerland. [3] Department of Environmental Systems Science, Institute of Terrestrial Ecosystems, ETH Zürich, Universitätstrasse 16, 8092 Zürich, Switzerland. [4] Terrestrial Ecology Research Group, Technische Universität München, Hans-Carl-von-Carlowitz-Platz 2, 85354 Freising, Germany. [5] Department of Biological Sciences, Middle East Technical University, 06800 Ankara, Turkey. [6] Field Station Fabrikschleichach, Department of Animal Ecology and Tropical Biology, University of Würzburg, 96181 Rauhenebrach, Germany. [7] Silviculture and Forest Ecology of the Temperate Zones, University of Göttingen, Büsgenweg 1, 37077 Göttingen, Germany. [✉]email: michael.staab1@tu-darmstadt.de

The decline of biodiversity is one of the most pressing ecological problems of the 21st century[1]. Recently, attention has focused on insects[2,3] where individual studies and a global meta-analysis indicated that insect populations in terrestrial ecosystems have declined over the last decades, but that trends differed widely among taxa, regions and ecosystems[4,5]. Until now, most scientific studies estimating long-term insect population changes were from agricultural landscapes and identified land use as potential driver[6–9], which stirred public and political debates. In contrast, changes in insect populations in forests have garnered comparably little attention (e.g[10]), even though forests cover over 30% of all terrestrial land and harbour a large share of the global species diversity[11]. In landscape mosaics, forests are often considered refuges for biodiversity buffering against negative effects in adjacent agricultural land[12,13]. Indeed, stable or increasing insect populations have been found in some forests, for example, for carabid beetles[12] and moth biomass[14]. Other studies, however, reported negative trends in forests across taxonomic groups[15–19].

Interpreting insect population trends in forests is challenging since population dynamics can be related to several short- and long-term processes that act simultaneously. Under natural conditions, stochastic disturbances and subsequent succession drive forest dynamics[20,21] and change local insect populations[10]. As most temperate forests are managed[22], human interventions can interfere with natural dynamics by altering habitat conditions and resource availability, e.g. by simplifying forest structure[23], potentially causing longer-term changes in insect communities[24]. To complicate the picture, climate change may amplify changes in forest structures and microclimate[17], even though the forest canopy partially buffers climatic conditions[25,26]. In addition to local site-scale drivers, there are drivers at the landscape scale, such as habitat composition in the surroundings[27] that can affect insects, for example through spill-over effects from agricultural land via pesticide drift[28] and nitrogen deposition[29,30] or through changed microclimate, in particular because forests are highly fragmented worldwide. To disentangle the many different potential mechanisms influencing insect populations in forests, detailed analyses are needed that consider both, changes in local site conditions as well as landscape effects. In managed forests, local site characteristics related to niche availability (tree species composition, canopy cover or deadwood volume) are often linked to forestry[31] and can thus potentially be targeted in forest management plans and conservation strategies to stabilize or increase insect populations. The landscape scale (forest cover and structure), may inform on relevant properties that usually can be targeted by landscape planning.

While analyses of insect abundance, biomass and species numbers are important to unravel overall changes, complementary trait-based approaches considering the life-history of species may allow to understand mechanisms[5]. In the context of insect decline, trait analyses on species-level have largely been applied to relatively well-studied taxa like Lepidoptera (butterflies and moths)[32]. These studies indicate that species with a narrow dietary niche and special habitat requirements declined disproportionally compared to generalists[5,6,15,32,33]. In addition to specialization, several other traits such as body size, trophic position, dispersal ability or commonness may interact with environmental or land-use variables in determining whether species gain or lose individuals over time. Body size is interweaved with many aspects of a species' biology. Larger species require more resources, making them more sensitive to changes in resource availability, and are usually less common[34]. Indeed, relationships between large body size and threat or decline have been shown for several insect taxa including moths[32], ground beetles[35] and saproxylic beetles[36]. Species decline can also be related to dispersal ability[19,36], possibly because species' ability to form meta-populations is related to dispersal[19]. Likewise, population

trends can be related to trophic level[35], as top consumers are more sensitive towards environmental perturbations[37,38], and as population sizes may be smaller at the top of the trophic pyramid[39]. Locally rarer species could be more prone to decline[40], as already the loss of relatively few individuals could infer with the probability of finding a reproduction partner, ultimately reducing local populations below viable sizes.

We present data from the Biodiversity Exploratories project[41], where beetles and true bugs were sampled and identified to species for 10 years at 140 forest sites across Germany, spanning a gradient in forest management from unmanaged broadleaf forests to intensively managed locally non-native conifer forests[19,41]. A previous study[19] indicated that species number and biomass of insects declined over time while drivers of the decline remained unclear, as neither a local composite land-use intensity index nor the proportion of arable fields in the surrounding landscape explained insect trends. Here, we aim to identify which environmental variables at the local (i.e. study site) and the landscape scale (i.e. forest around a study site) relate to this insect decline in forests by considering differences in conditions among sites (e.g. deadwood volume) as well as their site-specific temporal changes (e.g. change in deadwood volume), as both may determine population trends[42,43]. To quantify relationships at site and species level, we used Pearson´s r between sampling year and the respective community response measuring the correlation with time rather than the slope.

We hypothesize that insect decline at the site level is linked to management-induced changes in stand structure. Therefore, intensively managed forests may imply more consistent decline (i.e. negative correlation with time) in arthropod communities. At the landscape scale, we expect that low forest cover and low structural complexity of forests within a landscape will accelerate decline due to lower habitat availability. With respect to species' traits, we expect species with large body size and correspondingly smaller population size to decline more. Moreover, we hypothesize that declines ascend through the food web, with higher trophic levels (carnivores) having more negative correlations with time.

## Results

**Site-level**. Correlations per site were expressed as Pearson's r that measures the strength of the correlation between insect community responses and sampling year. Calculations were based on 120,996 individuals from 1,805 species (in 37,006 species × site × year combinations) from 140 sites, of which 30 sites were sampled yearly and 110 sites triennially, with no influence of sampling effort on correlations (Supplementary Fig. 3). From 2008 to 2017, correlations were on average negative for total species richness (mean Pearson's $r = -0.182$, 95% CI $-0.257 – -0.106$) and biomass ($-0.152$, $-0.237 – -0.066$), but not abundance ($0.011$, $-0.079 – 0.100$) (Fig. 1a–c and Supplementary Table 2).

We found negative statistical effects of the proportion of non-native trees on correlations (Pearson's r) for species richness (estimate $= -0.106 \pm 0.049$ SE, $p = 0.032$), abundance ($-0.177 \pm 0.050$, $p = 0.001$) and biomass ($-0.137 \pm 0.050$, $p = 0.016$), with correlations becoming more negative when sites were dominated by non-native trees (Fig. 2a–c) (full statistical details available in Supplementary Data 1). Species richness correlations were further negatively related to change in deadwood volume ($-0.073 \pm 0.036$, $p = 0.044$), change in the proportion of non-native trees ($-0.115 \pm 0.042$, $p = 0.008$), change in canopy openness ($-0.130 \pm 0.040$, $p = 0.002$), and positively to PC1 of landscape heterogeneity (PC1 of Sentinel-1 radar backscatter in 1000 m radius, $-0.130 \pm 0.040$, $p = 0.002$) (Fig. 3a–d). Although abundance correlations were not negative on average, they decreased with harvesting intensity ($-0.089 \pm 0.044$, $p = 0.045$) in addition

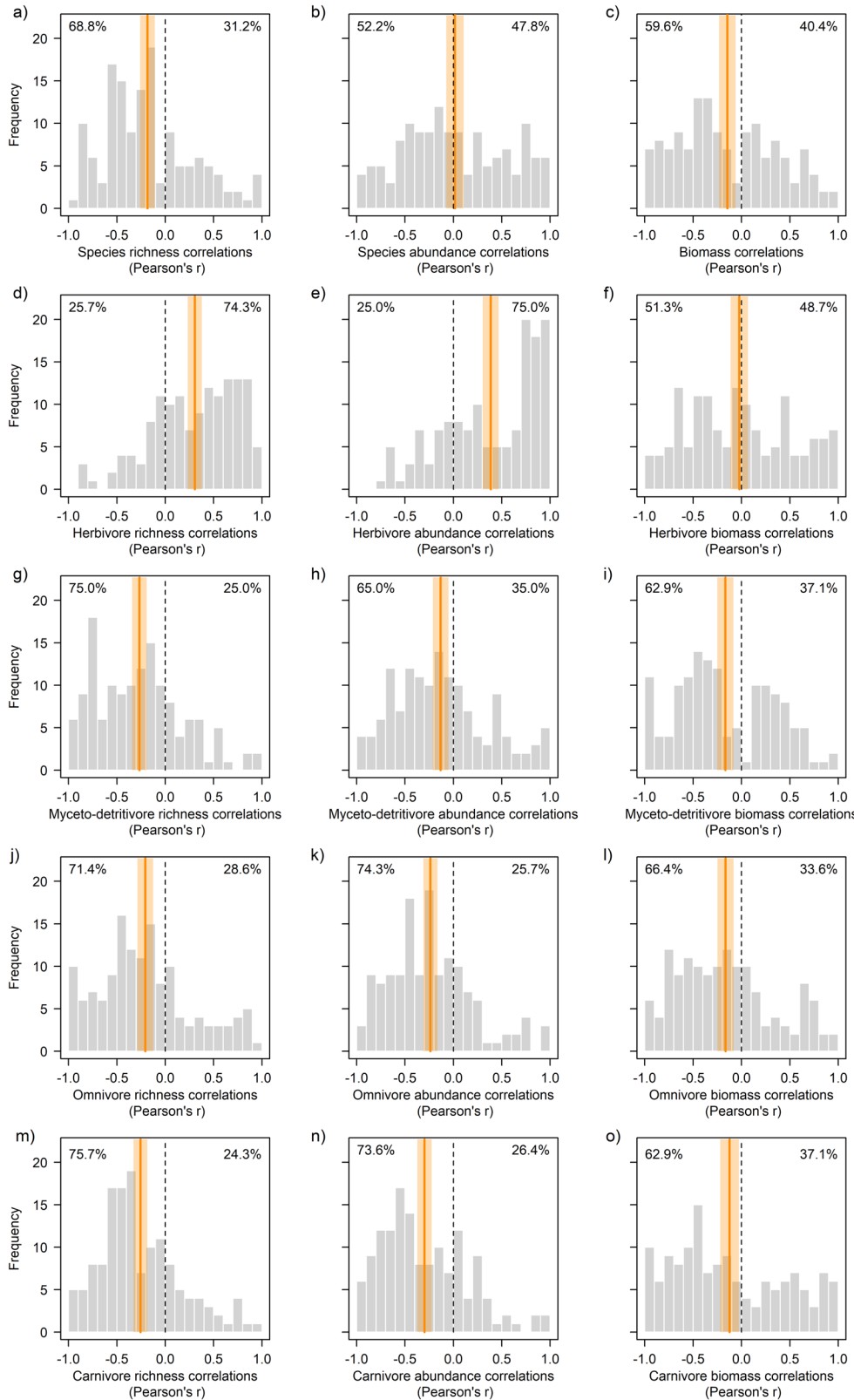

**Fig. 1 Distribution of site-level correlations.** Shown is Pearson's *r* between year and the respective community response in species richness (left column), abundance (middle column), and biomass (right column) for **a**–**c** the total insect community, **d**–**f** herbivores, **g**–**i** myceto-detritivores, **j**–**l** omnivores, and **m**–**o** carnivores. Bold vertical lines indicate average correlations (95% in shaded polygons). Correlations were negative on average with the exception of positive average correlations for herbivore species richness and abundance, while for total abundance and herbivore biomass 95% CI overlapped with null. Dashed vertical lines mark null with negative values indicating sites with declining and positive values sites with increasing community responses over time. Numerical details are reported in Supplementary Table 2. Percentages in each panel give the proportion of sites with negative (left) and positive (right) correlations.

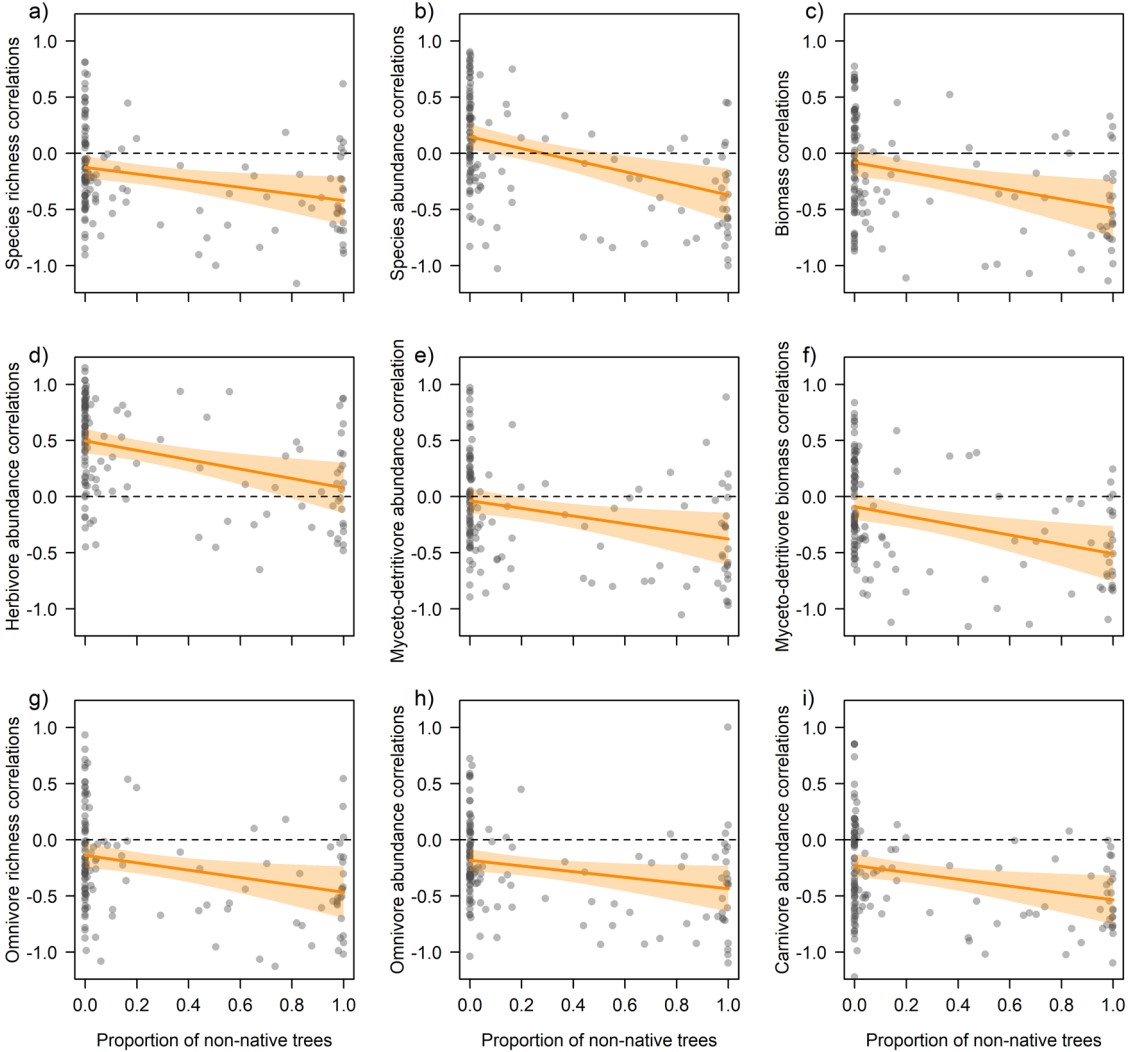

**Fig. 2 Results for site-level correlations and the proportion of non-native trees.** The proportion of non-native trees per site was related to site-level correlations (each shown as partial residuals of Pearson's *r* between year and the respective community response) in insect populations, with sites characterized by more non-native trees having more negative (or less positive) correlations in **a** total insect species richness, **b** total insect abundance, and **c** total insect biomass. The relationship with non-native trees was found for all trophic groups as indicated for **d** herbivore abundance, **e** myceto-detritivore abundance, **f** myceto-detritivore biomass, **g** omnivore species richness, **h** omnivore abundance, and **i** carnivore abundance. Note that even for **d** herbivores, which had positive correlations with time, abundance correlations on sites with more non-native trees were lower. Full statistical details are available in Supplementary Data 1. Regression lines (95% CI in shaded polygons) indicate the marginal predictions of linear mixed-effects models. Dashed horizontal lines mark null with negative values indicating sites with declining and positive values sites with increasing community responses over time.

to the decrease with proportion of non-native trees: at sites with little or no timber removal before the onset of sampling in 2008, abundance correlations were on average positive but became negative with more harvesting (Fig. 4a). Likewise, abundance correlations were negatively related to the effective number of layers (remotely-sensed vertical vegetation layering, ENL, −0.084 ± 0.042, *p* = 0.046) and to change in canopy openness (−0.104 ± 0.042, *p* = 0.014) but positively related to tree diversity (0.103 ± 0.043, *p* = 0.017) (Fig. 3e–g). Biomass correlations were not only related to the proportion of non-native trees but also to their change (−0.107 ± 0.044, *p* = 0.029, Fig. 3h). Sites that lost non-native trees had positive correlations that became negative towards sites that gained non-native trees. Once abundance was accounted for (see methods), no variable at the site or landscape scale explained species richness correlations.

Site-level correlations in species richness, abundance and biomass varied among trophic groups (Fig. 1d–o and Supplementary Table 2) but were nevertheless positively associated with each

other and with total correlations (Supplementary Fig. 11). Myceto-detritivores, omnivores, and carnivores had negative correlations with time on average, with carnivore abundance (mean Pearson's *r* = −0.299, 95% CI −0.371 – −0.223) and myceto-detritivore species richness (-0.272, -0.347 – -0.197) being most negative. In contrast, for herbivores, species richness (0.306, 0.232 – 0.379) and abundance (0.389, 0.304 – 0.468) correlations were on average positive, while herbivore biomass remained constant (-0.023, -0.112 – 0.067). Like for the total insect community, the proportion of non-native trees was related to correlations of different trophic groups (Supplementary Data 1). A negative relationship between site-level correlations and non-native trees prevailed, which was significant for herbivore abundance, myceto-detritivore abundance and biomass, omnivore species richness and abundance, and carnivore abundance (Fig. 2d–i). For herbivores, sites without non-native trees had predominantly positive correlations (Pearson's *r* > 0) in abundance that vanished towards sites dominated by non-native trees. In turn, for the other trophic groups, correlations were

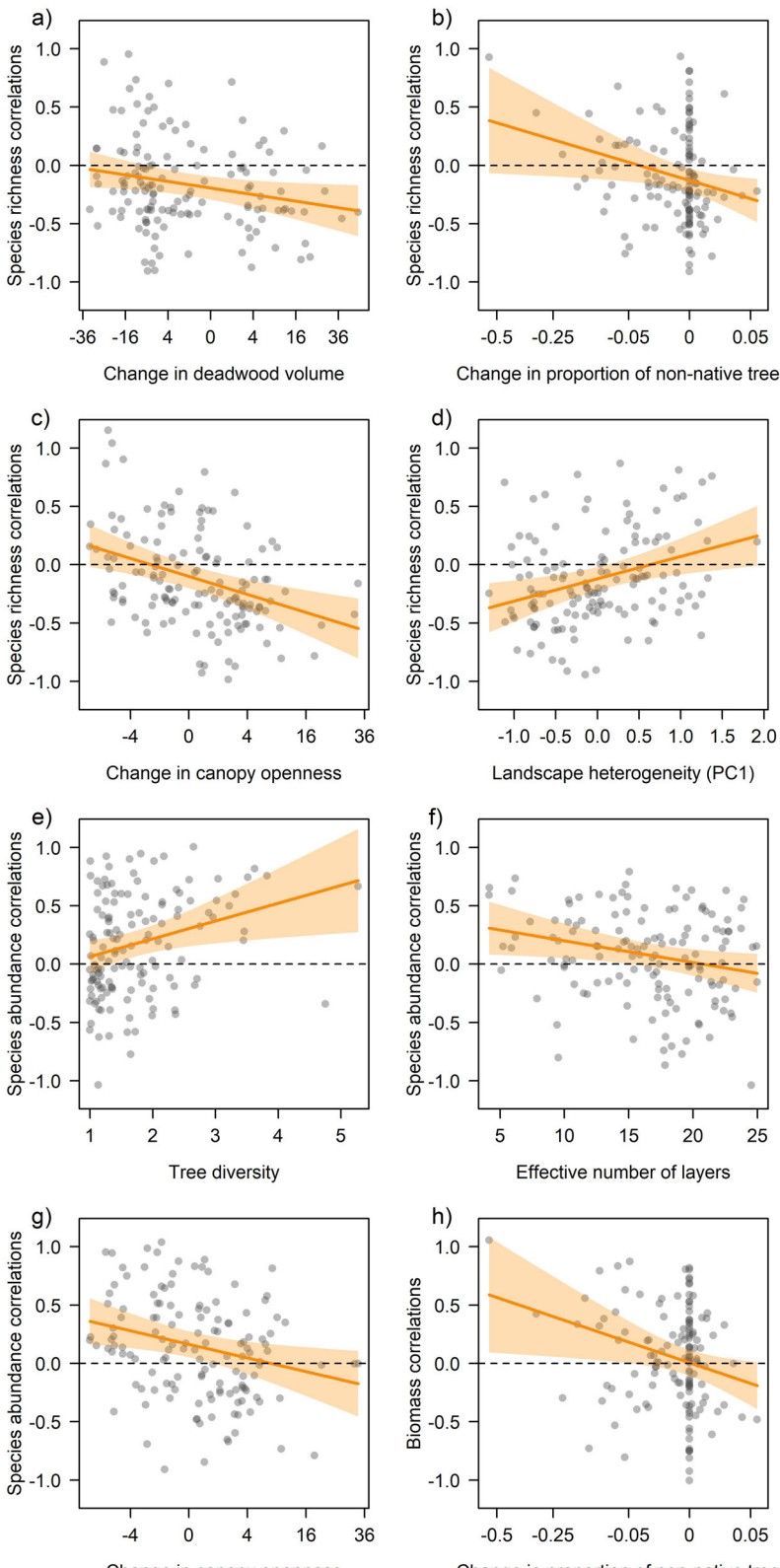

**Fig. 3 Results for site-level correlations of the total insect community and further environmental variables.** Site-level correlations (each shown as partial residuals of Pearson's *r* between year and the respective community response) for species richness were related to **a** change in deadwood volume, **b** change in the proportion of non-native trees, **c** change in canopy openness, and **d** PC1 of landscape heterogeneity. For species abundance, site-level correlations were related to **e** tree diversity, **f** the effective number of layers, and **g** change in canopy openness. Site-level correlations for biomass were related to **h** the change in the proportion of non-native trees. For explanations of variables see Supplementary Table 1. Full statistical details are available in Supplementary Data 1. Regression lines (95% CI in shaded polygons) indicate the marginal predictions of linear mixed-effects models. Dashed horizontal lines mark null with negative values indicating sites with declining and positive values sites with increasing community responses over time. Note that the *x*-axes in **a**, **b**, **c**, **g** and **h** are on a symmetric square-root scale.

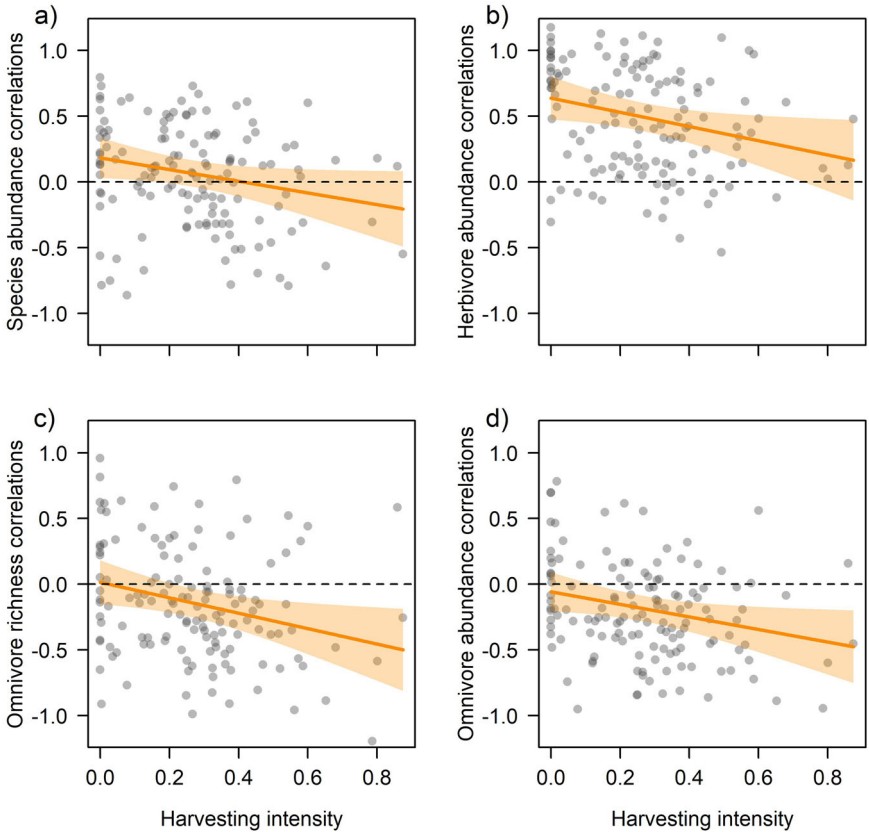

**Fig. 4 Results for site-level correlations and harvesting intensity.** Harvesting intensity before the start of insect sampling was related to site-level correlations (each shown as partial residuals of Pearson's *r* between year and the respective community response) in insect populations, with sites characterized by higher harvesting having more negative (or less positive) correlations in **a** total insect abundance. The relationship with harvesting intensity prevailed across trophic groups as indicated for **b** herbivore abundance, **c** omnivore species richness, and **d** omnivore abundance. Note that even for **b** herbivores, which had positive correlations with time, abundance correlations on sites with more harvesting were lower. Full statistical details are available in Supplementary Data 1. Regression lines (95% CI in shaded polygons) indicate the marginal predictions of linear mixed-effects models. Dashed horizontal lines mark null with negative values indicating sites with declining and positive values sites with increasing community responses over time.

largely negative (Pearson's $r < 0$) and got more negative at sites with a higher proportion of non-native trees. Similarly, most site-level correlations were negatively associated with harvesting intensity, significantly so for herbivore abundance and omnivore species richness and abundance (Fig. 4b–d).

Depending on the trophic group, further site- and landscape-scale variables influenced correlations between insect community responses and sampling year (Supplementary Figs. 7–10 and Supplementary Data 1), with most significant relationships found in omnivores and fewest in carnivores. Herbivore species richness correlations increased with change in harvesting (i.e. timber removal 2008–2017) and decreased with change in the proportion of non-native trees. Herbivore abundance and biomass correlations became more positive when landscape-scale disturbance intensity (percentage forest area in 1000 m radius affected by canopy-changing disturbance from 2008–2017) was high. For myceto-detritivores, species richness and biomass correlations were negatively related to change in the proportion of non-native trees. Myceto-detritivore richness correlations were, furthermore, negatively related to the change in canopy openness, while biomass correlations in this trophic group increased with tree diversity, being negative at low tree diversity and becoming positive at high tree diversity. Omnivore species richness correlations decreased with change in deadwood volume and change in canopy openness. Correlations of omnivore abundance and biomass were negatively associated to the latter and to change in the proportion of non-native trees. Omnivore biomass

correlations also decreased with landscape-scale forest cover and increased with disturbance intensity. Lastly, carnivore species richness correlations were negatively related to change in canopy openness. Accounting a priori for abundances indicated a few further nuanced relationships (Supplementary Data 1). Correlations of abundance-accounted species richness of herbivores were negatively related to canopy openness and change therein, while abundance-accounted species richness correlations of omnivores decreased with change in effective number of layers. If 2008 data were included as a covariate (Supplementary Data 2), all correlations were strongly negatively associated to the respective conditions in the first sampling year. While several results regarding environmental variables differed from the models reported above, significant associations with harvesting intensity did not change (Supplementary Data 2).

**Species-level**. Not only the majority of sites but also the majority of species had negative correlations (Pearson's *r* between year and the number of individuals per species in a region) with time (Supplementary Fig. 12). Following our expectation, correlations of individual species per region (based on 1050 species in 1874 species*region combinations, singletons excluded) were related to the species' traits (Supplementary Table 3). Larger species (Fig. 5a) had more consistently negative correlations with time (estimate $= -0.026 \pm 0.011$ SE, $p = 0.016$) as did more abundant species (Fig. 5b) ($-0.039 \pm 0.009$, $p < 0.001$), i.e. species

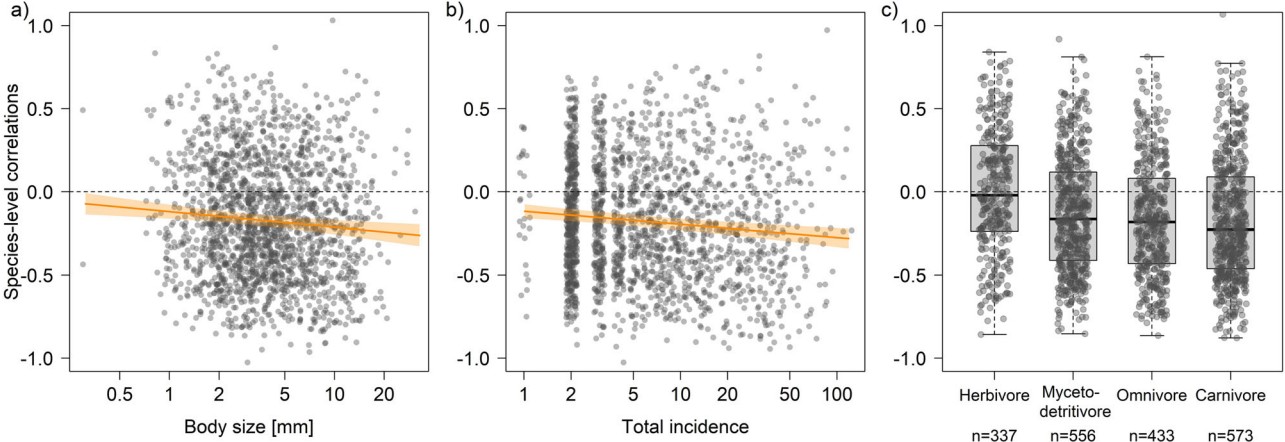

**Fig. 5 Results for species-level correlations and traits.** Species-level correlations decline with **a** body size and **b** total incidence of each species per region. Correlations (shown as partial residuals of Pearson's *r* between year and the number of individuals per species in a region, excluding single occurrences) varied among **c** trophic groups, with the majority of the species in all groups except herbivores having negative values on average. Numbers in **c** indicate species*region combination for each trophic group (center lines in boxplots specify the median, boxes cover the range between the lower and upper quartile, whiskers extend to 1.5x interquartile range). Statistical details are given in Supplementary Tables 3 and 4. Regression lines (95% CI in shaded polygons) indicate the marginal predictions of a linear mixed-effects model. Dashed horizontal lines mark null with negative values indicating species with declining and positive values species with increasing individual numbers over time.

which had a higher cumulative incidence across all years in a region. Similar to site-level, species-level correlations varied among trophic groups (Fig. 5c). On average, correlations for herbivores were slightly positive but distinctly negative for all other groups (Supplementary Fig. 12) that each had significantly different pairwise contrasts with herbivores (all *p* < 0.001, Supplementary Table 4). The largest difference was observed between herbivores and carnivores. In contrast to trophic group, species' correlations were not influenced by dispersal ability and stratum use preferences. When analysed separately for persistent (found in 6 or more years per region) or unsteady (found in 5 or fewer years per region) species, correlations differed. For unsteady species (1231 species × region combinations), results were qualitatively and quantitatively similar to the full data (Supplementary Tables 3 and 4), with significant relationships for body length, total individual numbers and trophic group. For the persistent species (643 species × region combinations), in turn, species' correlations were only related to trophic group, suggesting that species-specific correlations with year are particularly driven by less persistent species, even though the number of years a species was found did not influence correlations (Supplementary Fig. 13).

## Discussion

A spatially highly replicated time series shows that forest insects in three regions of Germany have declined in biomass and species richness, with the majority of species having shrinking populations. However, variation among sites, trophic groups and species (depending on their traits) was pronounced. The data add to the growing number of studies documenting declines in terrestrial arthropods across ecosystems [7,18,19,44–47]. We linked the correlation between sampling year and the respective insect community responses to static conditions and temporal changes in environmental variables, which have to our knowledge not been tested as moderators of insect decline in forests yet. Even though most statistical effect sizes were rather moderate and the time series spans only 10 years, which is just a fraction of the natural succession of any forest, our study goes beyond previous work in forests by identifying potential environmental drivers and by comprehensively linking species-level correlations to traits.

Forest management influences forest structure and resource availability, which may each shape insect populations and changes therein. Many studies have tested for the influence of forest structure on insect abundance and diversity (refs. [23,31,48–52]). For some taxa, diversity can be higher in intensively managed forests (refs. [31,49,51]), but old-growth forests harbour species absent from managed forests (refs. [52,53]). While previous studies primarily documented forest management effects on insect abundance and diversity, we focused on changes in insect communities over time. We found that the proportion of non-native trees, i.e. trees that would without human intervention not grow on the sampled sites (spruce and pine) was an important driver of decline in overall species richness, abundance and biomass and for at least one community response in every trophic group. This indicates that promoting native trees at the cost of non-natives, in Germany often conifers, can be one measure to halt insect declines, which is reinforced by the more negative correlations of species richness and biomass when the proportion of non-native trees had increased. As there were no significant results with non-native trees once abundance was accounted for, our inference points to a resource-driven mechanism, following species-energy theory and the more individuals hypothesis [54,55]. This also suggests that changes in site-level habitat heterogeneity [56] are not dominating correlations with time, or we would have found quantitively similar relationships for correlations of species richness with and without abundance accounted for [51].

In addition to non-native trees, prior harvesting intensity, i.e. the proportional amount of timber removed in the 10–15 years before the start of the insect time series, but not the change in harvesting during the sampling period, was associated with temporal changes in total insect abundance, as well as species richness of omnivores and abundance of herbivores and omnivores. As there was no large-scale shift in general environmental conditions and climate (*sensu*[10], data do not cover the drought years since 2018), an influence of forestry activities on changes in site-level insect populations is possible, which is also suggested by results with other environmental variables such as change in canopy openness. Mechanistically, trees that are not part of the natural species pool at a given site may have fewer associated consumer species or trophic links with primary consumers may be weaker[57]. Harvesting (including thinning) removes tree

individuals and may particularly in forests with few native trees lower resource availability provided by trees and alter habitat conditions and microclimate. Thus, it is not surprising that among the trophic groups, correlations of plant feeders (herbivores and omnivores), which together comprise 43% of species and 55% of individuals were negatively related to non-native trees and harvesting. Non-native trees in our sites are conifers, with a relatively open canopy fostering the growth of understory vegetation[58], which provides resources and habitat for insects preferring these conditions[51]. However, negative overall correlations prevailed. The underlying mechanisms remain unresolved, but reasons may include food web changes associated with a community turnover, reduced shelter and stronger exposition to natural enemies in more open forests[59]. Furthermore, once the canopy closes, the movement of ectothermic flying insects might be reduced subsequently lowering the chance that an individual hits a trap, which might explain the results with change in canopy openness. This could indicate that spots with open canopy not caused by tree removal, which would be the case in forests with natural dynamics, can contribute to halt declines. Notably, the only positive associations with environmental variables were with high tree diversity (abundance correlations) and a heterogenous forest landscape (species richness correlations), suggesting that high resource diversity across scales may be able to counteract insect declines.

There were only comparatively few relationships with landscape-scale variables. Species richness correlations of all insects were positively related to increasing structural heterogeneity (derived from satellite radar) of the surrounding forests. Against our expectation, correlations of the total community and of all trophic groups (except omnivore biomass) were independent of landscape-scale forest cover. At least for the study regions with comparably high amount of forest (mean forest cover in 1000 m radius = 81.0% ± 15.9 SD), this may suggest that conditions at the local site rather than in the surrounding landscape influence insect populations. Hence, forest insect communities in landscapes with high forest cover may be relatively resistant and resilient to impacts of adjacent agriculture[19] (but see[60]).

Nevertheless, our evidence is correlative. For establishing causality, experiments[61] that combine insect monitoring with the manipulation of single forest properties (such as tree species composition or harvesting) while keeping others constant would be desirable albeit logistically challenging. Because site selection in the Biodiversity Exploratories was stratified to include a wide gradient in land-use intensities[41], we can rule out a site selection bias arising when deliberately choosing monitoring sites with high species diversity[62]. As cyclic population dynamics in forest insects are expected to be multidecadal—if data from the few studied pest species are extrapolated to forest insects in general[63]—surveying our insect populations into the future will be necessary to fully understand how the environment and changes therein influence populations. This could also reveal if correlations with time were related to baseline insect species richness, abundance and biomass[64], which could be the case as results partially changed once conditions in the first sampling year were included as covariate. However, the likelihood of 'Regression to the mean' artifacts in short time series cautions against the ecological interpretation of models with 2008 conditions.

Solely focusing on the overall community might mask different responses of individual trophic groups or species with particular traits. Separating herbivores, myceto-detritivores, omnivores and carnivores revealed important relationships beyond non-native trees. Herbivores, the only group with on average increasing species richness and abundance (but not biomass) over time, gained species particularly at sites with more harvesting during the sampling period and individuals at sites with higher landscape-scale disturbance, counteracting negative influences of non-native trees and harvesting. Several of the herbivore species increasing most are either generalists or associated with European beech, the native tree species that was particularly favoured over conifers during the last decades in regional silviculture[65]. Thus, positive bottom-up effects of resource availability[66], also considering that disturbed landscapes are expected to have high herbaceous plant diversity, together with the close associations between herbivores and host plant species[67] may explain why this trophic group was bucking the negative total sign and the sign of all other trophic groups. Being released from enemies, who have declined most, may provide a further explanation.

Myceto-detritivores are important decomposers whose long-term declines in Central Europe are associated with historical changes in forest structures[53]. In our data, negative correlations with time were mostly related to non-native trees, which was for species richness attenuated by changes in canopy openness and for biomass partly counteracted by high tree diversity. Many myceto-detritivores may benefit from open forests[50] and the heterogenous microclimate therein, which may explain the observed temporal changes. Surprisingly, neither correlations of myceto-detritivores, many of which are saproxylic species that depend at least partly on deadwood[43], nor of any other trophic group were related to deadwood. This result reinforces that deadwood availability in most Central European forests—although increasing[68]—is still far below the quantities expected in old-growth forests with natural tree mortality dynamics[52,69], and is not yet sufficient to influence site-level insect correlations.

Top consumers are often most sensitive to environmental change[38]. Correlations in carnivore species richness, abundance and biomass were consistently negative and only sparsely associated to environmental variables at the site and landscape scale. As species-environment relationships may or may not ascent in food webs, this may blur the relationship between carnivores and the environment (see also ref. [70]), because their populations in forests are influenced by complex feedbacks between prey availability and environmental variables possibly hindering the detection of unequivocal drivers[48,71]. In summary, even though the statistical effect sizes in our analyses were rather moderate, our results are a first piece in solving the puzzle of declining insect populations in Central European forests. Nevertheless, there was congruence in correlations among all trophic groups (Supplementary Fig. 11), which indicates that population change may *en large* also be related to several drivers that act in concert[3] or by joint drivers not included in our analyses that act independently of the analysed site- and landscape-scale variables. Climate warming, by destabilizing trophic interactions[72], by exerting physiological stress[73], or by amplifying other factors may be one potential explanation[45].

An additional interesting finding was that site-level species richness and abundance correlations were across trophic groups frequently related to different environmental variables than biomass correlations. It was recently proposed that biomass is a suitable and resource-efficient surrogate for species diversity when studying insect decline (refs. [7,74]), but our results support studies questioning this simplistic view[3,75,76].

While general reductions in insect populations have brought 'insect decline' on the international agenda, it is always the individual species whose populations are responding to environmental changes. Even though some insect species gained individuals over time, the majority of species declined, which is similar to plant species across Germany, where also more species have been losing than gaining[77]. Thus, it is important to understand which species tend to decline, which can be achieved by analysing their traits[3,78] that can correspond to species'

susceptibility to decline and finally to extinction. Species at highest risk are often characterized by small population size, large body size and high trophic level[1,35,36,38,53,79]. These traits can also inherently be linked. For example, species in higher trophic levels typically occur at lower densities. In line with our expectations, we found that species-level decline (correlation between year and the number of individuals per species in a region) was more pronounced for larger species, for more abundant species (i.e. found at more sites per year), and accelerated with increasing trophic level. The average decline was distributed unimodally and not driven by few species with particularly negative correlations with time, as was recently revealed for vertebrate populations where few catastrophic declines distorted the average[80].

Large-bodied species may generally be more sensitive to perturbations affecting their populations[1,32,36,40]. In insects, larger species have on average fewer generations per time and larval development is slower than for small-bodied species[32,36], lowering their probability to replace lost individuals and to adequately respond to habitat changes. Many of the same interrelations apply to less abundant species. However, we found that more, not less, abundant species experienced stronger declines. The more negative correlations with time for more abundant species concur with data for a range of insect taxa[46,74,81], indicating that it is not only the rare species that decrease, but also the common ones. If population decline would be purely stochastic, more negative correlations in relatively rare species would have been expected, which was not the case. In vertebrates, from which predictions were originally derived, rarity itself is a context-dependent predictor of population change[40,82]. For fragmented landscapes it has been postulated that naturally common insect species may depend more on the maintenance of metapopulations[83], which might have been affected by large-scale environmental drivers that were not captured in the analyses. Alternatively, common species may initially have been more resilient to population declines than rare species, the latter of which have already declined relatively more before the start of our time series. As changes in populations are often a gradual process, common species may have accumulated an extinction dept and lag behind. Nevertheless, even though we quantified species-level abundance conservatively based on incidence, we cannot fully rule out that a 'Regression to the mean' effect contributes to the result for species-level correlations and abundance.

Species-level correlations were strongly linked to the trophic group a species is affiliated with. While herbivores increased, all other groups showed declining populations. Herbivore species may benefit from changes in tree composition (see above). Furthermore, in a warming climate, plant defence may be lowered and herbivore performance increased[84,85], which could foster population growth rates at least for generalists, thus selectively promoting herbivores but not other trophic groups. Carnivores are typically more sensitive to environmental change[38] and have disproportionate local extinction risk[60]. Concurrent with declines in omnivores and myceto-detritivores (i.e. decomposers), these changes in species' populations may shift food webs[86] and affect important ecosystem functions including nutrient cycling and pest control[87,88]. Such a reduction in ecosystem functions could be accentuated by climate uncertainty, which might even increase the probability of herbivore outbreaks[89].

Insects in German forests have been declining in the majority of sites and species across trophic groups (except herbivores). Temporal changes in insect populations are complex and can rarely be associated to a single environmental variable. A multitude of factors that are often inherently related and difficult to disentangle, recently named the "death by a thousand cuts"[3], may contribute to declines. Our results indicate that declines are at the site level linked to resources that could be fostered with a targeted forest management. Currently, forest areas with a natural tree

species composition that are not managed or managed at lower intensity than in the past are increasing in Central Europe, and various silvicultural systems potentially beneficial for insects such as retention forestry, continuous cover forestry and natural regeneration are being implemented. While those forests will take a long time to resemble primeval forests, they may still contribute to halt and ultimately reverse the at present predominating losses. Notably, decline at species level was not uniform but related to species' traits, particularly trophic group, likely transferring into changes in food webs. While we only begin to understand the extent and consequences of insect decline in forests, not least because changes in populations are often gradual and response may lag behind, consequences for the functioning of forest ecosystems merit our attention.

## Methods

**Study sites**. The study was conducted in three regions of Germany (Schwäbische Alb, Hainich-Dün, Schorfheide-Chorin) as part of the Biodiversity Exploratories[41] (Supplementary Fig. 1). Study regions differ in climate and topography, covering a broad range of conditions representative of temperate forests in Central Europe. Sites in the Schwäbische Alb in the southwest are at an elevation between 460-860 m a.s.l. with a mean annual temperature of 6–7 °C and a mean annual precipitation of 700–1000 mm. The Hainich-Dün region is in the geographic center of Germany at 285–550 m a.s.l. with 6.5–8 °C and 500–800 mm. Schorfheide-Chorin is in the northeastern German lowlands at 3–140 m a.s.l., and the warmest and driest of the three regions with 8–8.5 °C and 500–600 mm.

A total of 140 forest sites (49 in Schwäbische Alb, 50 in Hainich-Dün, 41 in Schorfheide-Chorin) were originally selected from a large pool of candidate sites by a stratified, random sampling procedure. Thus, the design rules out site-selection biases arising from deliberately choosing sites with high abundance or diversity at the beginning of a time series[62]. Each study site measures 100 m × 100 m (1 ha) and is embedded within larger management units (to minimize potential edge effects) that have been continuously forested for a long time with no recent change in land cover. Sites in each region include a broad range of forest types representative of the conditions per region and of forests in Germany[90]. Forests differ in management intensity, spanning a gradient from unmanaged beech forest to intensively managed pure stands of conifers (spruce and pine). The potential natural vegetation in all regions is a temperate broad-leaved forest dominated by European beech (*Fagus sylvatica*). European spruce (*Picea abies*) and Scots pine (*Pinus sylvestris*) are native in Central Europe but would be absent (spruce) or very rare (pine) in the study regions under natural conditions. In the Schwäbische Alb and Hainich-Dün, sites consist of European beech and European spruce, while in Schorfheide-Chorin sites are in beech and Scots pine forests. Forests are managed by the respective owners or tenants, with no influence of researchers in management decisions.

**Insect data**. Flying insects were sampled yearly from 2008–2017 with two flight-interception traps[91] (40 cm × 60 cm) per site (details in Supplementary Methods) in the Core Project Arthropods within the Biodiversity Exploratories. Thirty representative sites (Supplementary Fig. 3) were intensively-studied and sampled yearly. For logistical reasons, the remaining 110 sites were sampled triennially in 2008, 2011, 2014 and 2017. For analyses, annual and triennial data were combined, with sampling effort being accounted for (see below). Traps were placed at randomly selected site corners at ~1.5 m above the ground and operated from March to October (yearly sampled sites) or from March to July (remaining sites), with collection jars replaced in approximately monthly intervals. To harmonize sampling effort among yearly vs. triennially-sampled sites, all site-level analyses were based on three sampling rounds covering May, June and July, the main flight-activity period of insects in Central Europe[92].

Specimens were first sorted to order. All adults of Coleoptera and Heteroptera, the two higher taxa for which taxonomic expertise was available and that occurred in large numbers, were identified to species level by contracted taxonomists (see acknowledgements). Only 0.25% of adult specimens could not be identified to species and were excluded from analyses. For every species, *body length* (BL, accuracy 0.01 mm) was measured from the collected specimens or compiled from the literature[93]. Biomass was calculated from BL with the power function of Rogers et al.[94]: biomass (in g) = $(0.0305 \times BL^{2.62})/1000$. We note that the biomass formula used in Seibold et al.[19] contained an error and that all biomass values reported therein are tenfold overestimated; the results remain unaffected.

To further describe the life histories of insect species, we characterized the traits *trophic group*, *dispersal ability*, and vertical *stratum use* as outlined in[93] from published sources and cross-checked by taxonomic experts. Based on their main food source, each species was assigned as either 'herbivore', 'myceto-detritivore', 'omnivore' (i.e. feeding on more than one main food source) or 'carnivore'. Dispersal ability was characterized in five gradations (from low to high) based on flight ability and wing dimorphism (see ref. [19]). Vertical stratum use (categories: 'ground', 'herb layer', 'shrub and tree layer', 'unspecific') describes the main

vegetation layer in forests that a species usually uses as the response of insect communities to forest management may differ among strata[51]. While those categorical traits are a relatively coarse representation of species' niche space, they facilitate testing for differences among life histories. Furthermore, to quantify the commonness of a species in a region, the *total incidence* (cumulative incidence across all sites and years per region) was calculated.

**Site and landscape data**. We conducted forest inventories and applied terrestrial and airborne laser scanning (details in Supplementary Methods). At each 100 m × 100 m (1 ha) site, two inventories were conducted (2009–2011 and 2015–2016, ~6 years interval)[56]. In brief, trees >7 cm diameter at breast height were censused. For deadwood, items >25 cm (including stumps) were recorded at the whole 1 ha, while smaller deadwood (7–25 cm) was extrapolated from two transects. We derived several measures characterizing forest properties and management. *Harvesting intensity* was calculated as the basal area of timber harvested in the ~10–15 year period before the start of insect sampling divided by the total basal area of living trees and harvested timber (sensu[95]). *Change in harvesting* was expressed as the basal area of trees felled from 2008–2017 (i.e. insect sampling period)[56]. *Deadwood volume* was calculated as the volumetric sum of all deadwood. The *proportion of non-native trees* was expressed as the volume of living trees, harvested trees and deadwood that do not belong to the native tree species composition (mainly spruce and pine) divided by the total volume. Data from the first inventory characterized the conditions at the beginning of the insect time series. In turn, the difference between inventories described the change in deadwood volume and non-native trees (condition$_{second\ inventory}$ − condition$_{first\ inventory}$). Thus, positive change indicates a higher value in the second inventory (e.g. increase in deadwood volume). As insect abundance and diversity in forests are often related to *tree diversity*[66], we calculated the exponential Shannon index (e^H) of all trees >7 cm diameter.

Vertical forest structure was characterized with terrestrial laser scanning (LiDAR) in 2014 and 2019[96]. The *effective number of layers* (ENL) was calculated as the inverse Simpson index of filled horizontal 1 m layers[96]. Higher ENL indicates vertically more evenly layered vegetation, and may relate to insect abundance and diversity[50,97]. From the same scans, we calculated *canopy openness* as the percentage of sky pixels[98]. As for forest properties, we considered the conditions from the first scan and the change between scans (e.g. ENL$_{second\ scan}$ − ENL$_{first\ scan}$).

The landscape (1000 m radius) was assessed from satellite data. *Forest cover*, a surrogate of landscape-scale habitat availability, was measured from 2009 land cover data[19]. Land-use type (forest vs. grassland/agriculture) in Germany is subject to legislation and did not change in 2008–2017. To quantify landscape-scale changes in forests we calculated *disturbance intensity*, the percentage of forest (1000 m radius) with changed canopy from 2008–2017[99]. We furthermore assessed the *heterogeneity of forests* (1000 m radius) from satellite-borne radar (Sentinel-1). Backscatter intensities representing structural heterogeneity from 2016 were processed[100] and subjected to a principal component analysis (PCA). The first two PCs accounted for 76.7% of the total variation (PC1: 53.9%, PC2: 22.8%). Sentinel-1 is only available since 2016, precluding assessment of change.

**Data analyses**. All analyses were calculated in R version 4.0.1 (www.r-project.org). First, we tested whether changes in insect communities per site are related to forest properties at the site and landscape scale. Temporal changes of insect species richness, abundance and biomass for all 140 sites were calculated as Pearson's r (i.e. Pearson's product-moment correlation coefficient) of log-transformed raw data versus year (outcome for non-transformed data is highly similar and correlated with r > 0.95 in all cases, Supplementary Fig. 2). The correlation coefficient Pearson's r is the ratio between the covariance of two variables and the product of their standard deviations. In our analysis, r is a measure of the strength of correlation between insect community responses (species richness, abundance, biomass) and sampling year. Values can range between −1.0 and 1.0 (in case of perfect negative and positive correlation, respectively) with values below 0 indicating declines and values above 0 indicating increases in the respective insect community response over time. While numerous approaches to describe temporal changes in biodiversity have been postulated[101], every method has advantages and disadvantages, and there is intense and unresolved actual debate on the suitability of individual measures (see the controversy on the Living Planet Index[80,102]). Thus, to quantify the correlations between insect community responses and sampling year (rather than their magnitudes), we opted for the simple but intuitively interpretable Pearson's r, which is an established standard effect size in ecology and evolution[103]; Pearson's r of log-transformed data is more robust against deviations in single years in relatively short time series than other metrics based on slopes. For the 30 yearly sampled sites, correlations are based on 10 years of data (2008–2017) and for the 110 triennially-sampled sites on all years with data (2008, 2011, 2014, 2017). Sampling intensity did not influence site-level correlations, which are comparable between yearly and triennially-sampled sites (Supplementary Fig. 3).

We computed separate analyses for the correlations in species richness, abundance, and biomass per site, for all insects together as well as separately for each trophic group (herbivores, myceto-detritivores, omnivores, carnivores). Pearson's r was the response variables in linear mixed-effects models calculated with the R-package 'lme4'[104]. Fixed effects at the site scale (1 ha) (Supplementary

Table 1 and Supplementary Fig. 4) were *harvesting intensity*, *change in harvesting*, *deadwood volume*, *change in deadwood volume*, *proportion of non-native trees*, *change in proportion of non-native trees*, *tree diversity*, *ENL*, *change in ENL*, *canopy openness*, and *change in canopy openness*. By including both the initial values of harvesting, deadwood volume, the proportion of non-native trees, ENL and canopy openness as well as the change in these variables during the insect sampling period, the analyses allow to disentangle whether correlations between year and the respective insect community response are related to the initial conditions at a site or to changes in conditions. To increase normality and homoscedasticity, deadwood volume was square-root transformed before analyses. For the same reason, we transformed all change values that were calculated as the difference between second and first condition. To cope with negative numbers arising when the first condition was larger than the second condition, we used symmetric square-root transformation by separately transforming the absolute values of numbers smaller and larger than zero, and subsequently multiplying all transformed values with their original sign, thus preserving the sign of the data. At the landscape scale (1000 m radius), fixed effects were *forest cover*, *disturbance intensity* (square-root transformed), and *forest heterogeneity* (PC1 of Sentinel data, PC2 was excluded due to variance inflation). We repeated all site-level analyses by additionally including the initial *conditions in the first year* of the insect time series (2008; log-transformed) to account for starting conditions at each site[105]. We are aware that including the starting conditions may lead to statistical artifacts known as 'Regression to the mean' because when an extreme observation is made in the beginning (e.g. high species richness), the subsequent observations are likely to be closer to the mean of the site. These analyses are here provided for comparison but we refrain from ecologically interpreting the results.

All fixed effects were scaled to mean = 0 and SD = ±1 to allow direct comparison of effect sizes. Region was treated as random intercept to account for possible variation among the three study regions (Schwäbische Alb, Hainich-Dün, Schorfheide-Chorin). Models were weighted by sampling intensity (the number of years available for calculating site-level correlations) to recognize the higher effort and replication of the yearly compared to the triennially-sampled sites. Because samples with more individuals are expected to contain more species (more individuals hypothesis[54]), which may make it difficult to separate influences of forest properties on species richness from influences mediated by abundance, we also calculated analyses that accounted for abundance. For this, we regressed richness with abundance (each log-transformed, for all insects and per trophic group) in linear models separately for each site, calculated correlations (Pearson's r) with time based on standardized residuals, and used these abundance-accounted species richness correlations as response variable as described above. We chose this approach to account for abundance over a classical rarefaction, because of some site per year combinations with very few collected individuals among trophic groups. By rarefying to the lowest observed abundance, a lot of the ecologically meaningful variability of the data would be discharged.

In addition to using Pearson's r as response variables in linear mixed-effects models, we also conducted multi-level mixed effects meta-analysis with Pearson's r as effect size, environmental variables as moderators, sampling effort as weights, and region as random effect[106]. Furthermore, we calculated linear mixed-effects models, in which the response variables were Theil-Sen Slopes (median slope of all possible pairwise slopes)[107] between insect community responses and sampling year. All three analytical approaches yielded highly congruent outcomes (Supplementary Fig. 5), affirming the use of Pearson's r between year and insect community responses for our statistical analyses.

In a second step we tested whether population correlations per species were related to the species' traits. Individuals from all sites of a region were summed per year because most species were only recorded on few plots per year, resulting in a high number of zeros. For each species per region combination, the correlations over the sampling period from 2008 to 2017 (10 years for the 30 plots with annual sampling) were calculated as Pearson's r between year and the number of individuals per species in a region. This way, we quantify the strength in the correlations between regional individual numbers (populations size) per species and time. Equivalent to site-level data, values below or above 0 indicate, respectively, that individual numbers of a species in a region declined or increased over time. Species-level correlations are based on raw individual numbers instead of log-transformed data, as several species were not recorded in every year, prohibiting log-transformation. On species level, we only analysed the 30 intensively-studied sites for which yearly data covering the entire vegetation period from March to October are available. This approach ensures that in all years the entire flight period of all species is covered by the sampling. Singleton species (i.e. only one individual found per region across all years) were excluded. Coverage of species per region is very similar between the yearly and the triennial data (Supplementary Fig. 6). Likewise, incidences of species per region were tightly correlated between the yearly and triennial data (r = 0.88), but the yearly data are expected to be less sensitive to distortion due to fluctuations in species-specific individual numbers than the triennial data.

A linear mixed-effect model with species-level correlations per region as response variable was calculated to test for the influence of the traits *body length*, *trophic group*, *dispersal ability*, *stratum use*, and of the commonness measure *total incidence*. By using incidences instead of individual numbers as measure for commonness, potential artifacts stemming from 'Regression to the mean' should be less likely as only the presence of a species per site but not its number of individuals

is considered. The numerical fixed effects (body length, total incidence, each log-transformed) were scaled to mean = 0 and SD = ±1 to allow direct comparison of effect sizes. Region and species identity were treated as crossed random intercepts to account for possible variation among regions and species per region. The species-specific model was weighted by frequency of occurrence (i.e. the number of years each species was found per region) to recognize the higher confidence in correlations for species present in more years. Additionally, to directly test whether species-level correlations depend on the regular presence of species per region, we repeated the species-level analyses separately for *persistent* (i.e. species found in 6 or more years per region) and *unsteady* species (i.e. species found in 5 or fewer years). When models indicated significance in categorical fixed effects (body length, trophic group, dispersal ability), pairwise contrasts, Bonferroni-Holm-corrected for multiple comparisons, were calculated with the R package 'emmeans'[108].

Model selection was not applied to either site-level or species-level analyses. To check for independence among fixed effects, variance inflation factors were calculated (<2.5 in all cases once PC2 was excluded). Residuals of all models were inspected for assumptions of normality and homogeneity of variances, which were always met.

## Data availability

All raw data are available in the BExIS repository (https://www.bexis.uni-jena.de/). Accession numbers: 22007, 22008 (raw arthropod data); 31122 (arthropod traits); 25786 (land cover); 22786, 22846 (forest structure); 27826 (effective number of layers); 27828 (canopy openness); 24526, 24546 (deadwood).

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

## Acknowledgements

We thank M. Lutz, J. Bartezko, P. Freynhagen, I. Gallenberger, M. Türke, M. Lange, T. Kahl, E. Pasalic, E. Sperr and all student helpers for conducting field sampling and laboratory sorting; C. Seilwinder and R. Honecker for GIS work; C. Ammer, J. Bauhus, M. Ehbrecht, P.

Magdon, S. Seibold and C. Senf for contributing data; C. Ammer, J. Müller and S. Seibold for critical comments. We are deeply indebted to all taxonomic experts: Coleoptera were identified by E. Anton, B. Büche, M.-A. Fritze, F. Köhler, T. Kölkebeck, L. Schmidt and T. Wagner; Heteroptera were identified by M. M. Gossner, R. Heckmann, C. Morkel and F. Schmolke. We thank the managers of the three Exploratories (S. Gockel, M. Gorke, A. Hemp, K. Lorenzen, K. Reichel-Jung, S. Renner, M. Teuscher, J. Vogt, K. Wiesner, K. Wells) for their work in maintaining the plot and project infrastructure; C. Fischer, M. Gleisberg, J. Mangels and S. Pfeiffer for giving support through the central office, J. Nieschulze, A. Ostrowski and M. Owonibi for managing the central data base; and M. Fischer, K. E. Linsenmair, D. Hessenmöller, D. Prati, I. Schöning, F. Buscot, E.-D. Schulze and the late E. Kalko for their role in setting up the Biodiversity Exploratories project. We thank the administration of the Hainich national park, the UNESCO Biosphere Reserve Swabian Alb and the UNESCO Biosphere Reserve Schorfheide-Chorin as well as all land owners for the excellent collaboration. The work has been funded by the German Science Foundation in the DFG Priority Programme 1374 "Biodiversity- Exploratories". Field work permits were issued by the responsible state environmental offices of Baden-Württemberg, Thüringen, and Brandenburg.

## Author contributions

M.M.G, W.W.W, and N.B designed research; M.S., M.M.G, N.K.S., R.A., D.A., S.B., P.S., W.W.W., and N.B. performed research; M.S. analysed the data; and M.S. wrote the paper, with input from W.W.W., N.B., and all coauthors.

## Funding

## Competing interests

The authors declare no competing interests.
