## [Peer Review File · Communications Biology]

Reviewers' comments:

Reviewer #1 (Remarks to the Author):

This is a well-written and interesting manuscript detailing the species level trends over 10 years in German forests, using the data of the Biodiversity Exploratories. The authors report some of the declines to be associated with timber harvest and non-native trees, and found differences among functional groups. They ALSO report that the sites with the highest richness, abundance and biomass at the start of sampling declined most.

This is a nice finding, but is unfortunately a statistical artifact, stemming from 'Regression to the Mean'. This also casts doubt on their finding that the most abundant species declines most (although this is not necessarily related), and the authors need to thoroughly show that this is not the case. More details on this below. I am, frankly, a bit astounded that none of the 9 authors, some of whom are statistically very well versed, have picked up on this artifact, and present it as a genuine result.

A second issue I have is that the 'trait based approach' is rather superficial, despite the promise of mechanistic understanding (L80). The authors assign the species to different functional and morphological groups, but the hypotheses of why certain groups might be declining or increasing, and how this may relate to any of the explanatory variables are extremely meager.

I have some written some comments on details at the end of this review.

Regression to the Mean (RttM) is the simple fact that if an extreme observation is made (e.g. high or low species richness), the subsequent observations are likely to be closer to the mean of the site. Hence, sites with a high diversity, abundance or biomass observed at the start (2008) are most likely to show a declining trend, as subsequent observations will be lower. Likewise, sites with very low diversity will most likely show a positive trend. This is exactly what is evident in fig S6, and such a strong relationship in ecology should be a red flag for anyone. The value of the start year should thus never be included as explanatory variable in a regression. The good news is though, that thanks to the good experimental setup, we can exclude the possibility that the sites were chosen based on their high diversity (which would lead to an excess of RttM effects).

This also casts doubt on the result that the most abundant species showed the strongest declines. However, this must not be the case. The authors defined the most abundant species as those that had the highest numbers over the complete time series. The risk here is that these were also the species that had the highest abundances in 2008, and thus RttM could also potentially explain the declines of these most abundant species. The authors must rigorously show that this is not a RttM effect. (there is evidence that previously very common species are indeed declining, so it's quite possible that this is real).

Regarding the statistical approach, I see no real problem with the 2-stage model used here, with correlation coefficients used as inputs for a multiple regression, though it's uncommon.

I was a bit surprised about the way the species specific trends were calculated: by pooling all observations of each species per region. I assume with pooling you mean summing, which means that sites with the highest abundance of a species will have most influence on the trend, and I actually don't think this is a reliable way of calculating a general trend for a species. A better way would be a mixed effect model with abundance as dependent variable, time as independent, and site nested in region as random variable.

Minor comments:

L51: 'gained attention'. you mean 'garnered attention'

L83: you generalise the results of 1 study here as if it's a globally established fact. Please rephrase

L93: Why would less common species be more prone to declines?

L582: Relation between richness and abundance. This is a rather complicated method. Why not simply calculate rarefied richness? This is easily done in the vegan package, to any level of abundance you like (usually the lowest observed abundance is taken to rarefy to).

L595 'relied on' you can make this more straightforward by stating that you only analysed these sites.

Reviewer #2 (Remarks to the Author):

Summary

The work summarizes a study where the authors studied the effect of local and landscape issues of forest dwelling insects' decline (Coleoptera and true bugs) based on a long-term monitoring scheme. One of the major advantages of the work that use well the functional approach to evaluate the long-term effect of forest management on insect communities. The work identifies well which insect functional groups (e.g. large forest carnivores and decomposers) were the most sensitive and which one opposed the global declining trends (herbivores). The context of the evaluation of the detected patterns is really elegant, the authors discuss all aspects of the possible consequences with very relevant references. All the conclusions are original and relevant.

General comments

The manuscript (ms) submitted as an article to the journal. The scopes and the suggested outline of the ms suits well with the journal's mission and I'm sure that the ms will attract readers to the journal. In general, the language of the ms is decent, the structure and the concept of the paper is neat, however text and figures may need a minor updates. Although the ms is very-well built, the text is rather extent in some occasion, especially in the discussion and the methods. The author did not utilize the available dimension for tables and figures, thus the reader mostly trust on the written text. In addition, I think it will be a good idea to emphasize more the functional approach in ms, as a conceptual skeleton through the text. The journal allows to use 5 figures and tables altogether, but only two figures included in the ms, and the authors often cite the ESM material, which makes the reading quite distorted. Moreover, I felt that the comparison with agricultural management and grasslands is quite arbitrary, due to the fact that forest management has different dynamics in space and time than that of agricultural lands.

Recommendation

Based on the all issues mentioned below, I may suggest a minor revision of the manuscript.

Specific comments

Lines 78-94 - In this section, I may suggest to focus on the beetles and true bugs response, the lepidoptera (lines 81-85) and grassland arthropod (lines 87-90) response slightly distort the logical line of this part, I may suggest to remove them.

Lines 110-112 should write: "We hypothesize that insect decline at the site level in forests is linked to the management induced changes in stand structure. Therefore, intensively managed forests may imply more consistent decline in arthropods communities."

Line 139 - Fig S8 can be transferred into the manuscript as a regular figure.

Line 149 - Fig S7 can be transferred into the manuscript as a regular figure. In addition I think the figure can be reduced and include only the response of myceto-detritivores and carnivores. With this updates all aspects of the functional response should be include in figures of the ms.

Line 217 - Here I may suggest to focus on the references from the palearctic region thus, the item 44, Floren & Linsemair, 2005 as a case study for tropical rainforest should be removed from here.

Lines 317- 342 - Although I have no any major issue the message in this paragraph, but I found that is slightly do not match the logical skeleton of the discussion; without this section the whole discussion would remain untouched in terms of coherency. Without this section I felt a better focus on the functional traits. In addition, the discussed issues in this section is a little bit far from the results based on my reading. So, I may suggest that authors may think about to at least reduce this section, or remove it.

Lines 363 -365 - Although the ecology as a discipline is based on some logical speculation, I found these sentences a little bit outfit from the here and have not identified any connection with results of this study. Thus I may suggest to consider these sentences for removal.

Lines 368-370 - The same issue as directly above.

Line 397 - I found that the project has an excellent website, thus some direct website citation can improve the visibility of the project. I think this is the core project "arthropods", which can be cited and acronym here. In addition I found that the method section is quite extent, thus some reduction might be necessary. I may suggest that authors may use the citation for the various project phases which is available at the website and shrink/reduce the text to the minimum parsimonious information to the reader.

Lines 424-467 - Based on my previous suggestion for the reduction of the methods I may propose to reduce this section at least 25% and provide the most essential information about the insect sampling.

Lines 469-527 - The same as above with exception that I may suggest to reduce by 50% since most of the GIS approach as a good literature background, thus it is not necessary to provide complete description of the applied protocols and variables derived from GIS resources.

Lines 854 - 888 - Acknowledgement, other credits and declarations should follow the body of text (methods) directly, and the references should be followed by the figures and tables (if any).

Reviewer #3 (Remarks to the Author):

The authors present analyses on a 10-year dataset of insect richness, abundance, and biomass for many, many forest sites and estimate effects of local and landscape level predictors on these three outcomes. This is the result of a massive effort and is an extremely valuable contribution not just to the insect conservation literature to help understand insect declines, but also broader themes in ecology such as the scale at which different drivers operate, and also has clear implications for conservation and management in forest systems. The authors have also made their decisions through the methods transparent and the supplemental information is excellent for reporting most information readers would want to know, plus the data are publicly archived which is a huge contribution to a field with few examples of detailed, long-term, spatially replicated datasets.

My only substantial concern, and it is a big one, is that the authors treat correlations as trends which is statistically not justifiable and also really muddles the interpretation which shows through in some of the awkward phrasing in the discussion. Pearson's r is a measure of the strength of the relationship between two variables around a trend. For non-stationary time series, which is what the authors are dealing with, there is also a relationship between the trend (i.e. the slope) and the correlation. So, for example, this relationship:

$$y \sim N(\mu, \sigma)$$
$$\mu = 0.9 * x$$
$$\sigma = 1$$

will produce a stronger r than:

$$y \sim N(\mu, \sigma)$$

$$\mu = 0.5*x$$
$$\sigma = 1$$

even though the y variable is just as scattered around the trend (i.e. sigma is equal) in both cases. This is because they are non-stationary (i.e. if you detrended the time series, they would have the same correlation). Conversely,

$$y \sim N(\mu, \sigma)$$
$$\mu = 0.5*x$$
$$\sigma = 0.1$$

will in have a stronger correlation than

$$y \sim N(\mu, \sigma)$$
$$\mu = 0.5*x$$
$$\sigma = 1$$

because there is more spread around the trend in the latter case.

Now, don't get me wrong, r is a perfectly acceptable measure of an effect size but it should not be called a trend. However, it is also, to some extent, just a more sophisticated way of vote-counting. For example, on lines 125-127, the authors take the mean of correlations and for abundance the confidence interval overlaps zero. The interpretation here should not be that there is no trend, but rather that the strength of the relationship for positive and negative relationships with any trend balanced each other out (in other words, just as many tightly linked relationships above and below 0, or slightly more nuanced vote counting).

I see two ways the authors could correct this that would result in (presumably) minimal changes to the overall interpretation but in a way that is statistically justifiable. 1) Simply switch to actually using the trend (i.e. the slope) instead of the correlation. The authors state in the methods they did not due this because of potential bias in slopes for short time series because of individual year outliers, however, there are alternative measures of slopes (e.g. non-parametric approaches) that are relatively robust to these types of issues. For example, one I personally like is the Theil-Sen slope which is essentially a median slope of all possible pairwise slopes, but there are other approaches for short time series would work as well. 2) Take an alternative approach to the analysis that allows you to use r but would have a slightly different interpretation. Because r is a standard measure of effect size, this seems like a natural dataset to use for a multi-level mixed effects meta-analysis. The authors could treat each site as a separate study and combine them meta-analytically with random and fixed effects as used now, but then the overall outcome would be the pooled effect. Using this approach, there would still be estimates of the effect of various covariates (e.g. the local and landscape data), but the interpretation would be about the implied relationship between time population/community responses so most of the trend language could be retained. This would also open up the possibility to include sampling effort as a weight so all the data from the annual and triennial sites could be used and just weighted differently. There is of course a third option that is really clunky, which is to 3) change all mentions of 'trends' throughout the results, discussion, and figures to refer to 'strength of the relationship between year and population/community response' instead of trend, but that would make the interpretation really challenging. For example, Fig2a then becomes a measure of the effect of harvesting intensity on the strength and direction of the relationship between species abundance and year. I really do think that overall the data and framing of the paper are great, but these results are not trends and should be not described or interpreted as such.

One overarching stylistic comment is that the authors use 'e.g.' and 'for example' way, way more often than is necessary. When revising, just keep an eye out for this phrase because it appeared

enough to be distracting at some points. I get the temptation to clarify when citing something that it is just one example of many that could be included, but it is perfectly okay to not have an exhaustive set of references to back up a statement. I suggest the authors check every instance of a parenthetical citation beginning with "e.g." and remove that in most cases because it isn't really necessary. Similarly, in the methods, you don't really need to say "...following (##)" because if you cite a paper in the methods, that is the assumption as to why you are citing it.

In the discussion, many paragraphs start with a quick recap of the results and then run through various reasons why that relationship may or may not be apparent. I think the discussion would be a lot stronger if the authors inverted the paragraphs to start with the generalities or a topic sentence and then point to places where this generalization was supported or not by the results, and then why or why not. This would also help tie the paper to the broader field of research and help place it in context instead of making the discussion almost entirely about the current paper results.

Some line by line edits and suggestions follow.

Eliza Grames
University of Nevada Reno

Line 49: I don't think it is fair to say that most scientific studies have focused on agricultural landscapes; revise to make it clear you are talking about within the context of estimating long-term insect population trends.

Line 62: The claim that most forests are managed also seems to be a bit too broad. I assume this is true in temperate systems, but unlikely to be the case for large amounts of subtropical and tropical forest. At the very least, add a citation to back this up.

Lines 67-70: Also changes in microclimate in fragmented forests.

Lines 74-75: Again, you may want to clarify what types of forestry and in which systems this would be the case.

Line 82: Leps certainly haven't been studied comprehensively! Maybe "well-studied" would be a better term here.

Lines 78-96: This paragraph is pretty awkward and hard to tell exactly what the point of it is until we get to your predictions. I recommend rephrasing it in terms of the traits that have been examined (which leads nicely to your hypothesis paragraph) and systems in which they have been found to be important and why that may be. Right now, it reads like a condensed bullet list of different examples but is not really synthesized to get across the point that I think you are trying to make, which is that traits are important and that body size and population size are two important things to consider.

Line 90: Drivers of decline can't really be related to species colonization ability. Drivers are what they are (e.g. climate change, land use change, etc.) but species traits can affect how species respond to those drivers.

Line 98: How does one rigorously identify a species? Unless there are more details here, the species were either identified or not identified. Relatedly, I am surprised the taxonomic experts were not offered co-authorship for their contributions to the paper.

Line 114: The phrase "increase decline" is a bit confusing. Maybe "accelerate decline" or "increase the rate of decline" would be better phrases.

Lines 184-185: This is an example of where the results are not really correct because you cannot say

based on correlation that some species "declined stronger" than others. I did not highlight all of these because of my general comments, but this applies throughout the results.

Lines 361-364: This is a really neat finding and worth highlighting more in the discussion!

Line 391: I think the authors mean to say "...ultimately reverse the..." not inverse.

Line 408: I am assuming you mean that you had study sites of 100x100m within forests of varying size, but at first I interpreted this as meaning you had selected a lot of very small forest patches as sampling sites. Probably worth a quick clarification.

Line 414: Are 'managed pure stands of conifers' referring to pine plantations, or heavily managed forests that would naturally be mostly coniferous?

Line 431: Extra phrase "were attached" tacked on to the end of this sentence.

Line 465: If you haven't already, I recommend checking if there are big differences between the total individual number calculated as pooled across all years and sites, and total individual number calculated only from years sampled as part of the triennial sampling. I imagine if there are some species that just happen to be more common in the 30 annual sites and some of the years which weren't part of triennial sampling (e.g. 2009, 2010) just happened to be weird years for other reasons (e.g. weather) then you could get a bias just from that.

Line 580: There is either a missing or extra parenthesis and/or comma in this line.

Line 618: Suggest rewriting as "Model selection was not applied to either site-level or species-level analyses."

Fig 3c: I suggest the authors consider something like a beehive plot instead of a boxplot so you can see the spread of the underlying datapoints besides just the quantiles.

Reviewer #1 (Remarks to the Author):

This is a well-written and interesting manuscript detailing the species level trends over 10 years in German forests, using the data of the Biodiversity Exploratories. The authors report some of the declines to be associated with timber harvest and non-native trees, and found differences among functional groups. They ALSO report that the sites with the highest richness, abundance and biomass at the start of sampling declined most.

This is a nice finding, but is unfortunately a statistical artifact, stemming from 'Regression to the Mean'. This also casts doubt on their finding that the most abundant species declines most (although this is not necessarily related), and the authors need to thoroughly show that this is not the case. More details on this below. I am, frankly, a bit astounded that none of the 9 authors, some of whom are statistically very well versed, have picked up on this artifact, and present it as a genuine result.

Authors' response: We thank the reviewer for the time and effort and for the supportive and candid evaluation of the manuscript. Please see our detailed accounts below, where we respond to all comments and explain all changes made (highlighted by colored text in the main manuscript, note that the suggested rearrangements stretch across many parts of the text without altering most of the content). We also are grateful for drawing our attention to the analyses and the potential of a 'Regression to the mean effect'.

For the site-level analyses, we agree that it was wrong to treat the relationships with the starting conditions as an actual result. In the revised version, we no longer include the conditions of the starting year (2008) as a covariate and all inference is now solely based on analyses without the starting year, which yielded quantitatively and qualitatively similar results. The corresponding section in the discussion was deleted and the reader is now alerted multiple times to 'Regression to the mean' (L 299-300, 383-385, 560-565, 612-614). Nevertheless, we note that some researchers explicitly advocate to "always include the initial values of a variable as a predictor when calculating models of its change..." (Mazalla & Diekmann, 2022, *Journal of Vegetation Science* 33: e13117). Thus, we retain for the site-level analyses a table in the supplementary material (Table S4) reporting results with the 2008 conditions included. This will allow readers favouring the approach suggested by e.g. Mazalla & Diekmann (2022) a comparison of models with and without the starting conditions but we do not interpret the inference of these analyses (L 299-300, 560-565).

For the species-level analyses, we now also explicitly mention that a contribution of a 'Regression to the mean' effect to the result that more abundant species decline more cannot be fully excluded (L 383-385). To further avoid biases due to high individual counts of species and eventual 'Regression to the mean' effects, we no longer use the number of individuals across all sampling years as measure for species-specific abundance. In the revised version, species' abundance in all species-level analyses is determined based on the cumulative incidence of a species over sites and years (L 475). Measuring abundance based on incidence is, among other fields, common in ant ecology, where a conceptually similar problem is prevalent due to coloniality and eventual distorted high individual counts (Gotelli et al. 2011 *Myrmecological News* 15: 13-19). By using the incidence of a species (summed per region over sites*year), potential artifacts stemming from 'Regression to the mean' should be unlikely as only the presence of a species per site but not its number of individuals is taken into account (L 611-614). All revised analyses that are solely based on incidence support the initial result that more abundant species decline more (L 201-202).

A second issue I have is that the 'trait based approach' is rather superficial, despite the promise of mechanistic understanding (L80). The authors assign the species to different functional and morphological groups, but the hypotheses of why certain groups might be declining or increasing, and how this may relate to any of the explanatory variables are extremely meager.

I have written some comments on details at the end of this review.

Authors' response: Following this suggestion, we specified and extended the text on traits in the introduction (L 78-97) and discussion L (349-385).

Regression to the Mean (RttM) is the simple fact that if an extreme observation is made (e.g. high or low species richness), the subsequent observations are likely to be closer to the mean of the site. Hence, sites with a high diversity, abundance or biomass observed at the start (2008) are most likely to show a declining trend, as subsequent observations will be lower. Likewise, sites with very low diversity will most likely show a positive trend. This is exactly what is evident in fig S6, and such a strong relationship in ecology should be a red flag for anyone. The value of the start year should thus never be included as explanatory variable in a regression. The good news is though, that thanks to the good experimental setup, we can exclude the possibility that the sites were chosen based on their high diversity (which would lead to an excess of RttM effects).

Authors' response: We thank the reviewer for drawing our attention to the potential of 'Regression to the mean' effects. As explained in our detailed reply above, the condition of the first year (2008) is no longer contained as covariate in all site-level analyses and caveats addressing 'Regression to the mean' in discussion and methods (L 383-385, 560-565, 612-614). The previous Figure S6 and the discussion paragraph have been removed.

This also casts doubt on the result that the most abundant species showed the strongest declines. However, this must not be the case. The authors defined the most abundant species as those that had the highest numbers over the complete time series. The risk here is that these were also the species that had the highest abundances in 2008, and thus RttM could also potentially explain the declines of these most abundant species. The authors must rigorously show that this is not a RttM effect. (there is evidence that previously very common species are indeed declining, so it's quite possible that this is real).

Authors' response: Abundance is in the revised version no longer defined as the number of individuals summed over all sites and years per region. Instead, to be much more conservative and to make 'Regression to the mean' effects less likely, abundance is determined based on the cumulative incidence of a species over sites and years (see the more detailed accounts above). We also added a note that 'Regression to the mean' cannot be fully ruled out as an explanation why more abundant species decline more (L 383-385).

Regarding the statistical approach, I see no real problem with the 2-stage model used here, with correlation coefficients used as inputs for a multiple regression, though it's uncommon.

I was a bit surprised about the way the species specific trends were calculated: by pooling all observations of each species per region. I assume with pooling you mean summing, which means that sites with the highest abundance of a species will have most influence on the trend, and I actually don't think this is a reliable way of calculating a general trend for a species. A better way would be a mixed effect model with abundance as dependent variable, time as independent, and site nested in region as random variable.

Authors' response: Before starting the analyses of the data for a similar dataset from grasslands (still unpublished), we explored several approaches, actually starting with abundance trends (i.e. beta coefficients / slopes) suggested here, i.e. mixed effect models with abundance as response variable. However, beta coefficients or slopes on species level showed a huge variation, were unstable, and often models did not converge. This was likely due to the high number of zeros in the raw data (L 591-592), as the larger majority of species is only recorded on few plots per year, and also due to outliers in abundance. By aggregating data per species and year (separately for each of the three geographically separated regions) and by using Pearson's r we characterize the changes in regional populations, which is in our opinion suitable for the research questions. See also our comments for reviewer 3 on different approaches suggested. We now clarify the chosen approach further in the methods (L 594-597). Actually, the concerns discussed above regarding 'Regression to the mean' reinforce the choice of Pearson's r, as the correlation coefficient is less sensitive than slopes.

Minor comments:

L51: 'gained attention'. you mean 'garnered attention'

Authors' response: Changed (L 51)

L83: You generalise the results of 1 study here as if it's a globally established fact. Please rephrase

Authors' response: The sentence has been rephrased (L 82-83).

L93: Why would less common species be more prone to declines?

Authors' response: This prediction was (also) deduced in the context of body size, as larger species have, on average, usually smaller population sizes. Species with locally smaller populations may be more prone to decline, as already the loss of few individuals could e.g. infer with the probability of finding a reproduction partner, ultimately bringing the species below its minimum viable population size for the respective region (L 94-97). Nevertheless, we acknowledge that predictions regarding the interplay between commonness/rarity and risk of the decline are at present largely derived from vertebrates (Chichorro et al., 2019, Biological Conservation 237: 220-229) (added text in L 374-375).

L582: Relation between richness and abundance. This is a rather complicated method. Why not simple calculate rarefied richness? This is easily done in the vegan package, to any level of abundance you like (usually the lowest observed abundance is taken to rarefy to).

Authors' response: We agree that the chosen approach to account for the relation between species richness and abundance may at first not appear elegant. In the beginning, we tried a classical rarefaction exactly in the suggested way. Rarefaction, however, has with our data the problem that for some trophic groups in a few years only very few individuals were collected in a few plots. By rarefying to the lowest observed abundance (e.g. 1 individual only for carnivores in one plot and year), we would discharge a lot of the ecologically meaningful variability of the data. As a further alternative, we had also tested the coverage-based interpolation/extrapolation framework proposed by Chao et al. (2014, Ecological Monographs 84: 45-67) as implemented in the iNEXT package (Hsieh et al., 2016, Methods in Ecology and Evolution 7: 1451-1456). This produced extremely broad standard errors and confidence intervals for samples with few individuals. Thus, as a compromise we settled for regressing

richness with abundance (each log-transformed) before calculating correlation coefficients based on the standardized residuals of the regression. We now justify the chosen approach to account for the relation between richness and abundance more clearly in L 578-581.

L595 'relied on' you can make this more straightforward by stating that you only analysed these sites.

Authors' response: Changed (L 600-601).

Reviewer #2 (Remarks to the Author):

Summary

The work summarizes a study where the authors studied the effect of local and landscape issues of forest dwelling insects' decline (Coleoptera and true bugs) based on a long-term monitoring scheme. One of the major advantages of the work that use well the functional approach to evaluate the long-term effect of forest management on insect communities. The work identifies well which insect functional groups (e.g. large forest carnivores and decomposers) were the most sensitive and which one opposed the global declining trends (herbivores). The context of the evaluation of the detected patterns is really elegant, the authors discuss all aspects of the possible consequences with very relevant references. All the conclusions are original and relevant.

Authors' response: We thank you for the very constructive evaluation of our manuscript. The suggestions are highly relevant and have considerably strengthened the manuscript.

General comments

The manuscript (ms) submitted as an article to the journal. The scopes and the suggested outline of the ms suits well with the journal's mission and I'm sure that the ms will attract readers to the journal. In general, the language of the ms is decent, the structure and the concept of the paper is neat, however text and figures may need a minor updates. Although the ms is very-well built, the text is rather extent in some occasion, especially in the discussion and the methods. The author did not utilize the available dimension for tables and figures, thus the reader mostly trust on the written text. In addition, I think it will be a good idea to emphasize more the functional approach in ms, as a conceptual skeleton through the text. The journal allows to use 5 figures and tables altogether, but only two figures included in the ms, and the authors often cite the ESM material, which makes the reading quite distorted. Moreover, I felt that the comparison with agricultural management and grasslands is quite arbitrary, due to the fact that forest management has different dynamics in space and time than that of agricultural lands.

Recommendation

Based on the all issues mentioned below, I may suggest a minor revision of the manuscript.

Authors' response: Following the suggestions, we have considerably changed several sections of the manuscript: the text in methods and discussion was shortened; the revised manuscript includes 5 figures, several figures that were previously in the supplementary material are now contained in the main text (note that the exact content of some figures has slightly changed due to the changes in the analyses proposed by reviewer 1 and 3); the writing concerning traits has been extended in introduction and discussion. We fully acknowledge that forests have different dynamics, and we have considerably reduced the comparison with grasslands and agricultural management. Nevertheless, we think that it is warranted in the context of the research to set our results in few selected instances into the context of the available knowledge, which is so far mostly derived from open habitats including grasslands and agriculture. Please see our detailed replies below where we explain the changes in more detail and reference the exact line numbers of the changes. In the main manuscript all changes made are highlighted by colored text. Note that the rearrangements in response to reviewer suggestions stretch across many parts of the text without altering most of the content

Specific comments

Lines 78-94 - In this section, I may suggest to focus on the beetles and true bugs response, the lepidoptera (lines 81-85) and grassland arthropod (lines 87-90) response slightly distort the logical line of this part, I may suggest to remove them.

Authors' response: This paragraph is now rewritten and more focused (L 78-97). We have removed the text on grassland arthropods and extended the text on forest beetles (including further references), to harmonize the outline in the introduction with the studied insect groups. We would prefer to not restrict the writing exclusively to Coleoptera and Heteroptera, as this would considerably weaken the reasoning, because most studies adopting a functional approach in the context of insect decline were conducted on insects other than beetles and true bugs.

Lines 110-112 should write: "We hypothesize that insect decline at the site level in forests is linked to the management induced changes in stand structure. Therefore, intensively managed forests may imply more consistent decline in arthropods communities."

Authors' response: Changed (L 111-113).

Line 139 - Fig S8 can be transferred into the manuscript as a regular figure.

Authors' response: The information from the former Figure S8 is now in the main manuscript as Figure 3. Please note that the content of the figure has slightly changed due to the changes in the analyses proposed by reviewer 1.

Line 149 – Fig S7 can be transferred into the manuscript as a regular figure. In addition I think the figure can be reduced and include only the response of myceto-detritivores and carnivores. With this updates all aspects of the functional response should be include in figures of the ms.

Authors' response: The former Figure S7 has been merged with the former Figure 1 into the new Figure 1. We decided to include the histograms for all trophic groups into the figure to give a comprehensive overview across groups.

Line 217 - Here I may suggest to focus on the references from the palearctic region thus, the item 44, Floren & Linsemair, 2005 as a case study for tropical rainforest should be removed from here.

Authors' response: The study by Floren and Linsemair has been removed.

Lines 317- 342 – Although I have no any major issue the message in this paragraph, but I found that is slightly do not match the logical skeleton of the discussion; without this section the whole discussion would remain untouched in terms of coherency. Without this section I felt a better focus on the functional traits. In addition, the discussed issues in this section is a little bit far from the results based on my reading. So, I may suggest that authors may think about to at least reduce this section, or remove it.

Authors' response: As suggested, the entire paragraph was removed without replacement. See also the critical comment of reviewer 1 concerning this inference.

Lines 363 -365 – Although the ecology as a discipline is based on some logical speculation, I found these sentences a little bit outfit from the here and have not identified any connection with results of this study. Thus I may suggest to consider these sentences for removal.

Authors' response: We agree that this sentence is a bit speculative. Nevertheless, we think that the argument fits well into this paragraph and we would prefer to keep the writing (L 372-374), especially in light of the suggestion from reviewer 3 to extend this aspect of the discussion.

Lines 368-370 - The same issue as directly above.

Authors' response: Similar to the previous comment, we would prefer to keep the writing (L 378-381) for the same reasons.

Line 397 - I found that the project has an excellent website, thus some direct website citation can improve the visibility of the project. I think this is the core project "arthropods", which can be cited and acronym here. In addition I found that the method section is quite extent, thus some reduction might be necessary. I may suggest that authors may use the citation for the various project phases which is available at the website and shrink/reduce the text to the minimum parsimonious information to the reader.

Authors' response: Following the suggestion, we have added a quotation to the project website (L 417) and now also explicitly refer to the Core Project Arthropods (L 444-445). The method section has been reduced and many details are now contained in the Supplementary Methods.

Lines 424-467 - Based on my previous suggestion for the reduction of the methods I may propose to reduce this section at least 25% and provide the most essential information about the insect sampling.

Authors' response: The section "Insect data" has been reduced by 27% to 432 words (initially 592 words). Details on e.g. sampling are now contained in the Supplementary Methods.

Lines 469-527 - The same as above with exception that I may suggest to reduce by 50% since most of the GIS approach as a good literature background, thus it is not necessary to provide complete description of the applied protocols and variables derived from GIS resources.

Authors' response: The section "Site and landscape data" has been reduced to 484 words (initially 840 words). Many specifications and details of protocols (including GIS) are now contained in the Supplementary Methods. In particular, we shortened the text on remote sensing, which was cut by 53% (from 457 to 216 words).

Lines 854 – 888 – Acknowledgement, other credits and declarations should follow the body of text (methods) directly, and the references should be followed by the figures and tables (if any).

Authors' response: The submission was a transfer from a different journal in the portfolio of the same publisher and thus formatted according to the guidelines of the original journal. The revised version strictly follows the formatting guidelines of *Communications Biology* (<https://www.nature.com/documents/commsj-life-style-formatting-guide-accept.pdf>), where the Acknowledgements have to be placed after the references.

Reviewer #3 (Remarks to the Author):

The authors present analyses on a 10-year dataset of insect richness, abundance, and biomass for many, many forest sites and estimate effects of local and landscape level predictors on these three outcomes. This is the result of a massive effort and is an extremely valuable contribution not just to the insect conservation literature to help understand insect declines, but also broader themes in ecology such as the scale at which different drivers operate, and also has clear implications for conservation and management in forest systems. The authors have also made their decisions through the methods transparent and the supplemental information is excellent for reporting most information readers would want to know, plus the data are publicly archived which is a huge contribution to a field with few examples of detailed, long-term, spatially replicated datasets.

Authors' response: We thank the reviewer for the detailed and constructive assessment of the manuscript. We highly appreciate the acknowledgement of the large and public dataset. Once more data from subsequent years will become available (our team is still clearing a sizeable backlog due to almost 2 years of Corona shutdown), we will also make those publicly available. Please see our detailed accounts below, where we respond to all comments and explain all changes made (highlighted by colored text in the main manuscript, note that the suggested rearrangements stretch across many parts of the text without altering most of the content).

My only substantial concern, and it is a big one, is that the authors treat correlations as trends which is statistically not justifiable and also really muddles the interpretation which shows through in some of the awkward phrasing in the discussion. Pearson's r is a measure of the strength of the relationship between two variables around a trend. For non-stationary time series, which is what the authors are dealing with, there is also a relationship between the trend (i.e. the slope) and the correlation. So, for example, this relationship:

$$y \sim N(\mu, \sigma)$$
$$\mu = 0.9 * x$$
$$\sigma = 1$$

will produce a stronger r than:

$$y \sim N(\mu, \sigma)$$
$$\mu = 0.5 * x$$
$$\sigma = 1$$

even though the y variable is just as scattered around the trend (i.e. σ is equal) in both cases. This is because they are non-stationary (i.e. if you detrended the time series, they would have the same correlation). Conversely,

$$y \sim N(\mu, \sigma)$$
$$\mu = 0.5 * x$$
$$\sigma = 0.1$$

will have a stronger correlation than

$$y \sim N(\mu, \sigma)$$
$$\mu = 0.5 * x$$

$\sigma = 1$

because there is more spread around the trend in the latter case.

Authors' response: We agree that the terminology we used in the manuscript was partly not congruent with the chosen analytical approach. This is now, among other changes, resolved by not referring to "trends" anymore when writing about the results and their interpretation. Please see our replies to the next comment directly below. We are grateful for the illustrative case and for reminding us of the properties of σ and μ . It is of course true that for the same value of μ , a higher value of σ will result in a stronger correlation. In our opinion, due to the inherent relationship between σ and μ (based on relative variation, standardized by the mean, i.e. coefficient of variation $CV = \sigma / \mu$) Pearson's correlation coefficient r fulfils the desirable properties for quantifying the strength (and sign) of the relationship between year and the respective population/community response.

Now, don't get me wrong, r is a perfectly acceptable measure of an effect size but it should not be called a trend. However, it is also, to some extent, just a more sophisticated way of vote-counting. For example, on lines 125-127, the authors take the mean of correlations and for abundance the confidence interval overlaps zero. The interpretation here should not be that there is no trend, but rather that the strength of the relationship for positive and negative relationships with any trend balanced each other out (in other words, just as many tightly linked relationships above and below 0, or slightly more nuanced vote counting).

I see two ways the authors could correct this that would result in (presumably) minimal changes to the overall interpretation but in a way that is statistically justifiable. 1) Simply switch to actually using the trend (i.e. the slope) instead of the correlation. The authors state in the methods they did not do this because of potential bias in slopes for short time series because of individual year outliers, however, there are alternative measures of slopes (e.g. non-parametric approaches) that are relatively robust to these types of issues. For example, one I personally like is the Theil-Sen slope which is essentially a median slope of all possible pairwise slopes, but there are other approaches for short time series would work as well. 2) Take an alternative approach to the analysis that allows you to use r but would have a slightly different interpretation. Because r is a standard measure of effect size, this seems like a natural dataset to use for a multi-level mixed effects meta-analysis. The authors could treat each site as a separate study and combine them meta-analytically with random and fixed effects as used now, but then the overall outcome would be the pooled effect. Using this approach, there would still be estimates of the effect of various covariates (e.g. the local and landscape data), but the interpretation would be about the implied relationship between time population/community responses so most of the trend language could be retained. This would also open up the possibility to include sampling effort as a weight so all the data from the annual and triennial sites could be used and just weighted differently. There is of course a third option that is really clunky, which is to 3) change all mentions of 'trends' throughout the results, discussion, and figures to refer to 'strength of the relationship between year and population/community response' instead of trend, but that would make the interpretation really challenging. For example, Fig2a then becomes a measure of the effect of harvesting intensity on the strength and direction of the relationship between species abundance and year. I really do think that overall the data and framing of the paper are great, but these results are not trends and should be not described or interpreted as such.

Authors' response: We agree with this comment regarding the suitability of Pearson's r as a measure for the relationship between two variables. When conceiving the analyses for a similar dataset from grasslands (still unpublished), we compared the properties of several approaches

to quantify the relationship between the year of sampling and the respective response at level of sites and species. In the end, the decision to settle on Pearson's r was based on its simplicity, its intuitive interpretation, standardization (being bound between -1 to 1) and thus direct comparability, and its wide use as an effect size (L 528-531). Also, we think that the correlation coefficient r has the advantage that it measures the strength (and consistency) of a relationship.

In the context of our research questions, especially the property regarding consistency is desirable. Consider a hypothetical case, where the number of species on a site is changing in one direction over the years, which is possibly related to an environmental condition. When the temporal change in the number of species is more consistent, the absolute value of r will be closer to 1, and using r as a response in a subsequent analysis will potentially enable to find an association with the particular environmental variable. This will also be the case, when the change in species numbers is not large, as long it is consistently negative (or positive). In turn, when using some sort of slope or related measure, this consistent but relatively small change may have a small value, as the slope is more driven by the absolute change and single outliers. See also our response above addressing a comment of reviewer 1.

We totally agree that we should not label our inference as "trend". Throughout the manuscript (including all figures and tables) we refined the terminology to "correlation" (as Pearson's r is a correlation coefficient), extended the explanation of why we used r (L 514-531, 582-589) and added recurring notes on what the analyses quantify (L 109-110, 121-122, 196, 223-224), including figure legends. The word "trend" is now only used for general statements that are not directly referring to the results. So, we basically took the third suggested option, and think that this does not make the writing too clunky. For example, in the figures for the site-level analyses, we now label the axes with "correlations" and then provide the detailed explanation "Pearson's r between year and the respective community response" in the figure legend.

Thank you also for suggesting the alternatives in form of Theil-Sen slopes (which none of the authors was so far aware of) and meta-analyses. We have tried both alternatives (Theil-Sen slopes as response variable in linear mixed-effects models; multi-level mixed effects meta-analysis with r as effect size, site/landscape variables as moderators and sampling effort as weights) and learned that the quantitative outcome was very similar to our approach of using r as response variable in linear mixed-effects models. The correlation between the beta coefficients of all three analyses was very high ($r=0.98$ for meta-analysis vs. mixed model with Pearson's r). To share this insight with our readers, we have added the new Supplementary Figure S5, where we show the relationships among analyses based on Pearson's r , Theil-Sen slopes and multi-level mixed effects meta-analysis for the example of the relationship between total species richness and sampling year. The high congruence among the complementary analyses enforces our confidence in the suitability of our statistical analyses. We have added a corresponding note to the methods, L 582-589.

One overarching stylistic comment is that the authors use 'e.g.' and 'for example' way, way more often than is necessary. When revising, just keep an eye out for this phrase because it appeared enough to be distracting at some points. I get the temptation to clarify when citing something that it is just one example of many that could be included, but it is perfectly okay to not have an exhaustive set of references to back up a statement. I suggest the authors check every instance of a parenthetical citation beginning with "e.g." and remove that in most cases because it isn't really necessary. Similarly, in the methods, you don't really need to say "...following (##)" because if you cite a paper in the methods, that is the assumption as to why you are citing it.

Authors' response: Almost all appearances of “e.g.” and “for example” (and related phrases) have been deleted. Likewise, phrases including “following” have been pruned from the generally shortened methods.

In the discussion, many paragraphs start with a quick recap of the results and then run through various reasons why that relationship may or may not be apparent. I think the discussion would be a lot stronger if the authors inverted the paragraphs to start with the generalities or a topic sentence and then point to places where this generalization was supported or not by the results, and then why or why not. This would also help tie the paper to the broader field of research and help place it in context instead of making the discussion almost entirely about the current paper results.

Authors' response: When revising the text, we also took care to restructure the paragraphs in the discussion. Topical sentences have been included (L 232-234, 303-304, 349-351).

Some line by line edits and suggestions follow.

Eliza Grames
University of Nevada Reno

Line 49: I don't think it is fair to say that most scientific studies have focused on agricultural landscapes; revise to make it clear you are talking about within the context of estimating long-term insect population trends.

Authors' response: Changed (L 49).

Line 62: The claim that most forests are managed also seems to be a bit too broad. I assume this is true in temperate systems, but unlikely to be the case for large amounts of subtropical and tropical forest. At the very least, add a citation to back this up.

Authors' response: We now restrict this claim to “temperate” forests, which is backed by a citation (L 61).

Lines 67-70: Also changes in microclimate in fragmented forests.

Authors' response: Microclimate has been added in L 69.

Lines 74-75: Again, you may want to clarify what types of forestry and in which systems this would be the case.

Authors' response: The text in L 72-74 has been adapted.

Line 82: Leps certainly haven't been studied comprehensively! Maybe “well-studied” would be a better term here.

Authors' response: “Comprehensively-studied” has been replaced by “relatively well-studied” (L 81).

Lines 78-96: This paragraph is pretty awkward and hard to tell exactly what the point of it is until we get to your predictions. I recommend rephrasing it in terms of the traits that have been examined

(which leads nicely to your hypothesis paragraph) and systems in which they have been found to be important and why that may be. Right now, it reads like a condensed bullet list of different examples but is not really synthesized to get across the point that I think you are trying to make, which is that traits are important and that body size and population size are two important things to consider.

Authors' response: Following the suggestion (and similar suggestions by reviewer 1 and 2) the paragraph has been completely restructured and rewritten (L 78-97).

Line 90: Drivers of decline can't really be related to species colonization ability. Drivers are what they are (e.g. climate change, land use change, etc.) but species traits can affect how species respond to those drivers.

Authors' response: The sentence is rewritten and does not refer to "drivers" anymore.

Line 98: How does one rigorously identify a species? Unless there are more details here, the species were either identified or not identified. Relatedly, I am surprised the taxonomic experts were not offered co-authorship for their contributions to the paper.

Authors' response: The word "rigorously" has been deleted. We would also like to take the opportunity to clarify the role of the taxonomic experts in the study: all individuals that identified specimens were paid contractors (L 457), with whom we have been working together since a long time (also in other projects). We would never exploit taxonomic expertise without a fair remuneration.

Line 114: The phrase "increase decline" is a bit confusing. Maybe "accelerate decline" or "increase the rate of decline" would be better phrases.

Authors' response: Changed, "accelerate" fits much better here (L 115).

Lines 184-185: This is an example of where the results are not really correct because you cannot say based on correlation that some species "declined stronger" than others. I did not highlight all of these because of my general comments, but this applies throughout the results.

Authors' response: We critically screened the text for such phrases and rewrote them. In this specific case we now write "had more consistently negative correlations with time" (L 200).

Lines 361-364: This is a really neat finding and worth highlighting more in the discussion!

Authors' response: We agree that the more consistent negative correlations for more common species is an interesting result. We have extended the corresponding discussion (L 369-384) that is necessarily a bit speculative. Nevertheless, we would prefer to not extend the writing even further, because reviewer 2 found this section too speculative and even suggested to remove it completely.

Line 391: I think the authors mean to say "...ultimately reverse the..." not inverse.

Authors' response: Changed, we of course mean "reverse" (L 407).

Line 408: I am assuming you mean that you had study sites of 100x100m within forests of varying size, but at first I interpreted this as meaning you had selected a lot of very small forest patches as sampling sites. Probably worth a quick clarification.

Authors' response: Yes, the text has been clarified (L 429-431).

Line 414: Are 'managed pure stands of conifers' referring to pine plantations, or heavily managed forests that would naturally be mostly coniferous?

Authors' response: Conifers would naturally be very rare, as the potential natural vegetation on all sites is a broad-leaved forest dominated by beech (*Fagus sylvatica*). This information was included in the same paragraph and has now been moved directly after the sentence commented on (L 434-438).

Line 431: Extra phrase "were attached" tacked on to the end of this sentence.

Authors' response: The error has been removed. Note that according to the suggestion by reviewer 2 this paragraph was condensed and partly transferred to the Supporting Information.

Line 465: If you haven't already, I recommend checking if there are big differences between the total individual number calculated as pooled across all years and sites, and total individual number calculated only from years sampled as part of the triennial sampling. I imagine if there are some species that just happen to be more common in the 30 annual sites and some of the years which weren't part of triennial sampling (e.g. 2009, 2010) just happened to be weird years for other reasons (e.g. weather) then you could get a bias just from that.

Authors' response: We had checked for congruence between yearly and triennial data (see also the related Venn Diagram, which was in Figure S5 and is now in Figure S6), which were correlated ($r=0.84$). Following the criticism by reviewer 1, we do no longer use individual numbers but incidences as measure of abundance (L 475). Species' incidences between yearly and triennial data are also correlated ($r=0.88$) and this information is added in L 606-607.

Line 580: There is either a missing or extra parenthesis and/or comma in this line.

Authors' response: This was an error. The missing parenthesis has been added (L 572).

Line 618: Suggest rewriting as "Model selection was not applied to either site-level or species-level analyses."

Authors' response: Changed according to the suggestion (L 627).

Fig 3c: I suggest the authors consider something like a beehive plot instead of a boxplot so you can see the spread of the underlying datapoints besides just the quantiles.

Authors' response: All boxplots now contain the underlying data.

REVIEWERS' COMMENTS:

Reviewer #2 (Remarks to the Author):

Authors have made an excellent revision with an accurately detailed replies for all issues raised by referees. In my opinion the manuscript request no further revisions before publication.

Reviewer #3 (Remarks to the Author):

The authors have done an excellent job addressing and incorporating reviewer comments and suggestions in the revised manuscript, and I continue to think it is an extremely valuable contribution to the literature on insect biodiversity change. The dataset itself is a huge contribution, and the authors have done a good job at being transparent with decisions in the analysis when reporting the results (e.g. reporting exact p-values, standard errors, non-significant findings, etc) and using adequate caution when interpreting the correlations. I especially like that the authors added two new approaches to the analyses (Sen slopes and meta-analyses) and found quantitatively similar results suggesting that the findings are relatively robust to different analytical choices and not simply an artefact.

Another strength of the study design is that sites were chosen with stratified random sampling, which eliminates some of the site selection bias often introduced in ecological studies (i.e. when researchers select sites that represent good habitat or have minimal disturbance such as nature preserves) as has been the case with several long-term insect biodiversity change studies. These types of data are essential to get a better picture of what is happening with insect populations and communities not just in relatively pristine or natural lands, but also semi-natural, managed, or degraded lands which comprise a large portion of available land.

I appreciate that the additions to the discussion on forest management (near lines 250-280) have clear implications for conservation practitioners and how forests can be managed to support insect biodiversity. I wonder if the authors could expand slightly on the changes in regional silviculture that produced positive trends for some insects (lines 309-313) because that seems like a conservation strategy that may be effective if it is already being adopted for silviculture.

Lines 58-59: Maybe "...since population dynamics can be related to several short..." would make more sense here? It seems strange to say populations themselves are related to processes.

Line 380: missing 'of' ("...the latter which have...")

Line 417: I would disagree with Reviewer #2 here about including the website link because it is not permanent. For example, the domain could change, the landing page for the English version could change, etc.

Reviewer #2 (Remarks to the Author):

Authors have made an excellent revision with an accurately detailed replies for all issues raised by referees. In my opinion the manuscript requests no further revisions before publication.

Authors' response: We thank you again for the very constructive evaluation of our manuscript.

Reviewer #3 (Remarks to the Author):

The authors have done an excellent job addressing and incorporating reviewer comments and suggestions in the revised manuscript, and I continue to think it is an extremely valuable contribution to the literature on insect biodiversity change. The dataset itself is a huge contribution, and the authors have done a good job at being transparent with decisions in the analysis when reporting the results (e.g. reporting exact p-values, standard errors, non-significant findings, etc) and using adequate caution when interpreting the correlations. I especially like that the authors added two new approaches to the analyses (Sen slopes and meta-analyses) and found quantitatively similar results suggesting that the findings are relatively robust to different analytical choices and not simply an artefact.

Another strength of the study design is that sites were chosen with stratified random sampling, which eliminates some of the site selection bias often introduced in ecological studies (i.e. when researchers select sites that represent good habitat or have minimal disturbance such as nature preserves) as has been the case with several long-term insect biodiversity change studies. These types of data are essential to get a better picture of what is happening with insect populations and communities not just in relatively pristine or natural lands, but also semi-natural, managed, or degraded lands which comprise a large portion of available land.

Authors' response: We thank you again for the detailed and thorough review.

I appreciate that the additions to the discussion on forest management (near lines 250-280) have clear implications for conservation practitioners and how forests can be managed to support insect biodiversity. I wonder if the authors could expand slightly on the changes in regional silviculture that produced positive trends for some insects (lines 309-313) because that seems like a conservation strategy that may be effective if it is already being adopted for silviculture.

Authors' response: We added examples of silvicultural systems that are at present being implemented and are potentially beneficial for insects in the conclusion (L 406-408).

Lines 58-59: Maybe '...since population dynamics can be related to several short...' would make more sense here? It seems strange to say populations themselves are related to processes.

Authors' response: Changed according to suggestion (L 58-59).

Line 380: missing 'of' ("...the latter which have...")

Authors' response: Revised to "the latter of which" (L 380).

Line 417: I would disagree with Reviewer #2 here about including the website link because it is not permanent. For example, the domain could change, the landing page for the English version could change, etc.

Authors' response: We fully agree. The link has been removed (L 419).